# Mapping the ecological networks of microbial communities

Yandong Xiao[1,2], Marco Tulio Angulo [3,4], Jonathan Friedman[5], Matthew K. Waldor[6,7], Scott T. Weiss[1] & Yang-Yu Liu [1,8]

Mapping the ecological networks of microbial communities is a necessary step toward understanding their assembly rules and predicting their temporal behavior. However, existing methods require assuming a particular population dynamics model, which is not known a priori. Moreover, those methods require fitting longitudinal abundance data, which are often not informative enough for reliable inference. To overcome these limitations, here we develop a new method based on steady-state abundance data. Our method can infer the network topology and inter-taxa interaction types without assuming any particular population dynamics model. Additionally, when the population dynamics is assumed to follow the classic Generalized Lotka–Volterra model, our method can infer the inter-taxa interaction strengths and intrinsic growth rates. We systematically validate our method using simulated data, and then apply it to four experimental data sets. Our method represents a key step towards reliable modeling of complex, real-world microbial communities, such as the human gut microbiota.

[1] Channing Division of Network Medicine, Brigham and Women's Hospital and Harvard Medical School, Boston, MA 02115, USA. [2] Science and Technology on Information Systems Engineering Laboratory, National University of Defense Technology, Changsha, Hunan 410073, China. [3] Institute of Mathematics, Universidad Nacional Autónoma de México, Juriquilla 76230, Mexico. [4] National Council for Science and Technology (CONACyT), Mexico City 03940, Mexico. [5] Physics of Living Systems, Department of Physics, Massachusetts Institute of Technology, Cambridge, MA 02139, USA. [6] Division of Infectious Diseases, Brigham and Women's Hospital and Harvard Medical School, Boston, MA 02115, USA. [7] Howard Hughes Medical Institute, Boston, MA 02115, USA. [8] Center for Cancer Systems Biology, Dana-Farber Cancer Institute, Boston, MA 02115, USA. Correspondence and requests for materials should be addressed to Y.-Y.L. (email: yyl@channing.harvard.edu)

The microbial communities established in animals, plants, soils, oceans, and virtually every ecological niche on Earth perform vital functions for maintaining the health of the associated ecosystems[1–5]. Recently, our knowledge of the organismal composition and metabolic functions of diverse microbial communities has markedly increased, due to advances in DNA sequencing and metagenomics[6]. However, our understanding of the underlying ecological networks of these diverse microbial communities lagged behind[7]. Mapping the structure of those ecological networks and developing ecosystem-wide dynamic models will be important for a variety of applications[8], from predicting the outcome of community alterations and the effects of perturbations[9], to the engineering of complex microbial communities[7,10]. We emphasize that the ecological network discussed here is a directed, signed, and weighted graph, where nodes represent microbial taxa and edges represent direct ecological interactions (e.g., parasitism, commensalism, mutualism, amensalism, or competition) between different taxa. This is fundamentally different from the correlation-based association or co-occurrence network[7,11–13], which is undirected and does not encode any causal relations or direct ecological interactions, and hence cannot be used to faithfully predict the dynamic behavior of microbial communities.

To date, existing methods for inferring the ecological networks of microbial communities are based on temporal abundance data, i.e., the abundance time-series of each taxon in the microbial community[14–19]. The success of those methods has been impaired by at least one of the following two fundamental limitations. First, those inference methods typically require the a priori choice of a parameterized population dynamics model for the microbial community. This choice is hard to justify, given that microbial taxa in the microbial community interact via a multitude of different mechanisms[7,20–22], producing complex dynamics even at the scale of two taxa[23,24]. Any deviation of the chosen model from the 'true' model of the microbial community can lead to systematic inference errors, regardless of the inference method that is used[19]. Second, a successful temporal data-based inference requires sufficiently informative time-series data[19,25]. For many host-associated microbial communities, such as the human gut microbiota, the available temporal data are often poorly informative. This is due to the fact that such microbial communities often display stability and resilience[26,27], which leads to measurements containing largely their steady-state behavior. For microbial communities such as the human gut microbiota, trying to improve the informativeness of temporal data is challenging and even ethically questionable, as it requires applying drastic and frequent perturbations to the microbial community, with unknown effects on the host.

To circumvent the above fundamental limitations of inference methods based on temporal data, here we developed a new method based on steady-state data, which does not require any external perturbations. The basic idea is as follows. Briefly, if we assume that the net ecological impact of species on each other is context-independent, then comparing equilibria (i.e., steady-state samples) consisting of different subsets of species would allow us to infer the interaction types. For example, if one steady-state sample differs from another only by addition of one species X, and adding X brings down the absolute abundance of species Y, then we can conclude X inhibits the growth of Y. This very simple idea can actually be extended to more complicated cases where steady-state samples differ from each other by more than one species. Indeed, we rigorously proved that, if we collect enough independent steady states of the microbial community, it is possible to infer the microbial interaction types (positive, negative, and neutral interactions) and the structure of the ecological network, without requiring any population dynamics model. We further derived a rigorous criterion to check if the steady-state data from a microbial community is consistent with the generalized Lotka–Volterra (GLV) model[15–19], a classic population dynamics model for microbial communities in human bodies, soils, and lakes. We proved that, if the microbial community follows the GLV dynamics, then the steady-state data can be used to accurately infer the model parameters, i.e., inter-taxa interaction strengths and intrinsic growth rates. We validated our inference method using simulated data generated from various classic population dynamics models. Finally, we applied it to real data collected from four different synthetic microbial communities, finding that the inferred ecological networks either agree well with the ground truth or help us predict the response of systems to perturbations. Our method represents a key step toward reliable ecological modeling of complex microbial communities, such as the human gut microbiota.

## Results

**Theoretical basis.** Microbes typically do not exist in isolation, but form complex ecological networks[7]. The ecological network of a microbial community is encoded in its population dynamics, which can be described by a set of ordinary differential equations (ODEs):

$$\mathrm{d}x_i(t)/\mathrm{d}t = x_i(t)f_i(\boldsymbol{x}(t)), \quad i = 1, \dots, N. \tag{1}$$

Here, $f_i(\boldsymbol{x}(t))$'s are some unspecified functions whose functional forms determine the structure of the underlying ecological network; $\boldsymbol{x}(t) = (x_1(t), \dots, x_N(t))^{\mathrm{T}} \in \mathbb{R}^N$ is an $N$-dimensional vector with $x_i(t)$ denoting the absolute abundance of the $i$-th taxon at time $t$. In this work, we do not require 'taxon' to have a particular taxonomic ranking, as long as the resulting abundance profiles are distinct enough across all the collected samples. Indeed, we can group microbes by species, genus, family, or just operational taxonomic units (OTUs).

Note that in the right-hand side of Eq. (1), we explicitly factor out $x_i$ to emphasize that (i) without external perturbations those initially absent or later extinct taxa will never be present in the microbial community again as time goes by, which is a natural feature of population dynamics (in the absence of taxon invasion or migration); (ii) there is a trivial steady state where all taxa are absent; (iii) there are many non-trivial steady states with different taxa collections. We assume that the steady-state samples collected in a data set $\mathcal{X}$ correspond to those non-trivial steady states $\boldsymbol{x}^*$ of Eq. (1), which satisfy $x_i^* f_i(x_1^*, \dots, x_N^*) = 0$, $i = 1, \dots, N$. For many host-associated microbial communities, e.g., the human gut microbiota, those cross-sectional samples collected from different individuals contain quite different collections of taxa (up to the taxonomic level of phylum binned from OTUs)[26]. We will show later that the number of independent steady-state samples is crucial for inferring the ecological network.

Mathematically, the intra- and inter-taxa ecological interactions (i.e., promotion, inhibition, or neutral) are encoded by the Jacobian matrix $J(\boldsymbol{x}(t)) \in \mathbb{R}^{N \times N}$ with matrix elements $J_{ij}(\boldsymbol{x}(t)) \equiv \partial f_i(\boldsymbol{x}(t))/\partial x_j$. The condition $J_{ij}(\boldsymbol{x}(t)) > 0$ (<0 or =0) means that taxon $j$ promotes (inhibits or does not affect) the growth of taxon $i$, respectively. The diagonal terms $J_{ii}(\boldsymbol{x}(t))$ represent intra-taxa interactions. Note that $J_{ij}(\boldsymbol{x}(t))$ might depend on the abundance of many other taxa beyond $i$ and $j$ (due to the so-called 'higher-order' interactions[24,28–32]).

The structure of the ecological network is represented by the zero-pattern of $J(\boldsymbol{x}(t))$. Under a very mild assumption that $\int_0^1 J_{ij}(\boldsymbol{x}^I + \sigma(\boldsymbol{x}^K - \boldsymbol{x}^I))\mathrm{d}\sigma = 0$ holds if and only if $J_{ij} \equiv 0$ (where $\boldsymbol{x}^I$ and $\boldsymbol{x}^K$ are two steady-state samples sharing taxon $i$), we find that the steady-state samples can be used to infer the

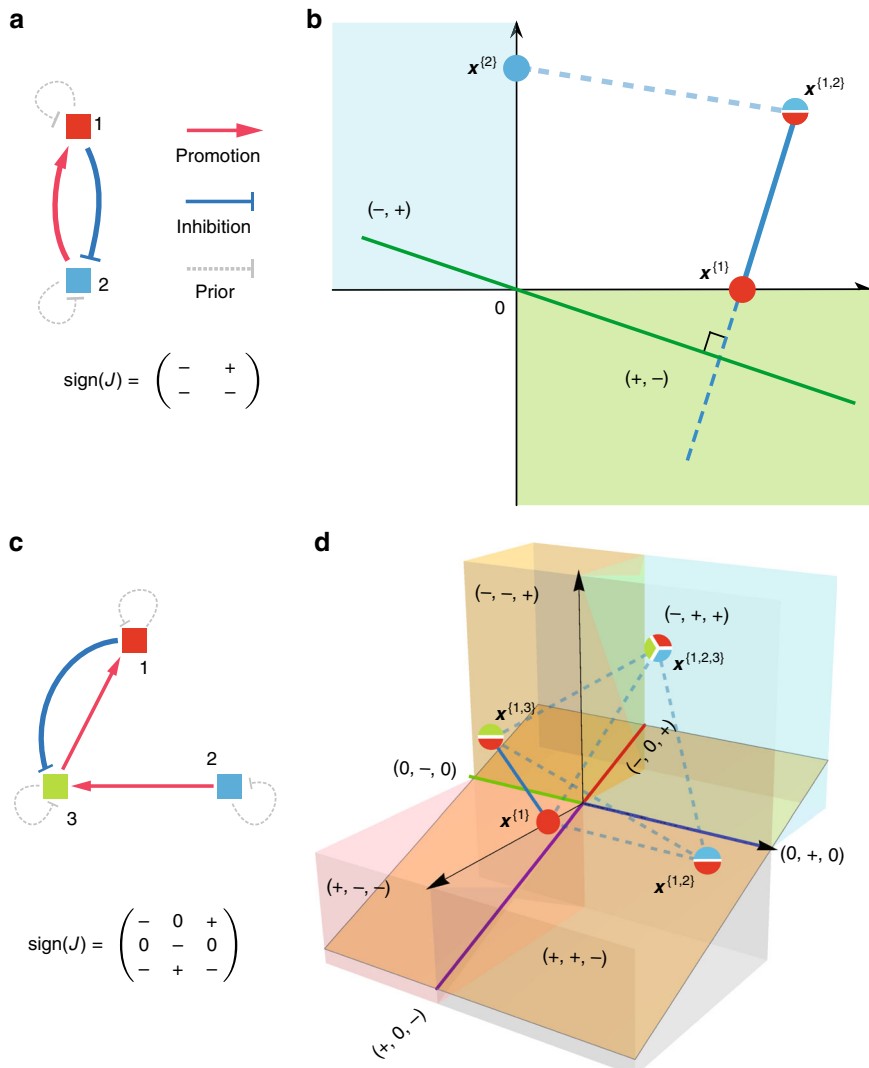

**Fig. 1** Inferring ecological interaction types for a small microbial community. The interaction types are coded as the sign-pattern of the Jacobian matrix. **a** For a microbial community of two taxa, its ecological network and the sign-pattern of the corresponding Jacobian matrix are shown here. **b** There are three possible steady-state samples (shown as colored pie charts), and two of them $x^{\{1,2\}}, x^{\{1\}}$ share taxon 1. We calculate the green line that passes through the origin and is perpendicular to the vector $(x^{\{1,2\}} - x^{\{1\}})$ (shown as a blue line segment). This green line crosses the origin, and two other orthants (shown in light cyan and green), offering a set of possible sign-patterns: $(0, 0), (-, -)$ and $(+, -)$, for which $s_1 = (\text{sign}(J_{11}), \text{sign}(J_{12}))$ may belong to. Provided that $J_{11} < 0$, we conclude that $s_1 = (-, +)$. **c** For a microbial community of three taxa, its ecological network and the sign-pattern of the corresponding Jacobian matrix are shown here. **d** There are seven possible steady-state samples, and we plot four of them that share taxon 1. Consider a line segment $(x^{\{1,3\}} - x^{\{1\}})$ (solid blue). We calculate the orange plane that passes through the origin and is perpendicular to this solid blue line. This orange plane crosses nine regions: the origin and the other eight regions (denoted in different color cubes, color lines), offering nine possible sign-patterns for $s_1$. We can consider another line segment that connects two steady-state samples sharing taxon 1, say, $x^{\{1,3\}}$ and $x^{\{1,2,3\}}$, and repeat the above procedure. We do this for all the sample pairs (dashed blue lines), record the regions crossed by the corresponding orthogonal planes. Finally, the intersection of the regions crossed by all those orthogonal hyperplanes yields a minimum set of sign-patterns $\hat{S}_1 = \{(-, 0, +), (0, 0, 0), (+, 0, -)\}$ that $s_1$ may belong to. If we know that $J_{11} < 0$, then we can uniquely determine $s_1 = (-, 0, +)$

zero-pattern of $J(x(t))$, i.e., the structure of the ecological network (Methods, the subsection 'Inferring the zero-pattern' of Supplementary Note 1, Supplementary Note 2, and Supplementary Note 3 for details). Note that the network structure is interesting by itself and can be very useful in control theoretical analysis of microbial communities[33]. But in many cases, we are more interested in inferring the interaction types or strengths, so that we can better predict the community's response to perturbations.

The ecological interaction types are encoded in the sign-pattern of $J(x(t))$, denoted as $\text{sign}(J(x(t)))$. To infer the interaction types, i.e., $\text{sign}(J(x(t)))$, we make an explicit assumption that $\text{sign}(J(x(t))) = \text{const}$ across all the observed steady-state samples. In other words, the nature of the ecological interactions

between any two taxa does not vary across all the observed steady-state samples, though their interaction strengths might change. Note that the magnitude of $J_{ij}(x(t))$ by definition may vary over different states, we just assume its sign remains invariant across all the observed samples/states. This assumption might be violated if those steady-state samples were collected from the microbial community under drastically different environmental conditions (e.g., nutrient availability[34]). In that case, inferring the interaction types becomes an ill-defined problem, since we have a 'moving target' and different subsets of steady-state samples may offer totally different answers. Notably, as we will show later, the assumption is valid for many classic population dynamics models[35–39].

The assumption that $\text{sign}(J(\boldsymbol{x}(t))) = \text{const}$ can be falsified by analyzing steady-state samples. In Proposition 1 of the subsection 'Inferring the sign-pattern' of Supplementary Note 1, we rigorously proved that if $\text{sign}(J(\boldsymbol{x}(t))) = \text{const}$, then true multi-stability does not exist. Equivalently, if a microbial community displays true multi-stability, then $\text{sign}(J(\boldsymbol{x}(t))) \neq \text{const}$. Here, a community of $N$ taxa displays true multi-stability if there exists a subset of $M (\leq N)$ taxa that has multiple different steady states, where all the $M$ taxa have positive abundances and the other $(N - M)$ taxa are absent. In practice, we can detect the presence of true multi-stability by examining the collected steady-state samples. If yes, then we know immediately that our assumption that $\text{sign}(J(\boldsymbol{x}(t))) = \text{const}$ is invalid and we should only infer the zero-pattern of $J$, i.e., the structure of the ecological network. If no, then at least our assumption is consistent with the collected steady-state samples, and we can use our method to infer $\text{sign}(J(\boldsymbol{x}(t)))$, i.e., the ecological interaction types. In short, by introducing a criterion to falsify our assumption, we significantly enhance the applicability of our method (see Remark 6 of the subsection 'Inferring the sign-pattern' of Supplementary Note 1 for more detailed discussions).

**Inferring interaction types**. The assumption that $\text{sign}(J(\boldsymbol{x}(t))) = \text{const}$ enables us to mathematically prove that $\text{sign}(J(\boldsymbol{x}(t)))$ satisfies a strong constraint (see Methods and Theorem 2 in the subsection 'Inferring the sign-pattern' of Supplementary Note 1). By collecting enough independent steady-state samples, we can solve for the sign-pattern of $J(\boldsymbol{x})$, and hence map the structure of the ecological network (Remarks 4 and 5 in the subsection 'Inferring the sign-pattern' of Supplementary Note 1).

The basic idea is as follows. Let $\mathcal{I}_i$ be the set of all steady-state samples sharing taxon $i$. Then, for any two of those samples $\boldsymbol{x}^I$ and $\boldsymbol{x}^K$, where the superscripts $I, K \in \mathcal{I}_i$ denote the collections of present taxa in those samples, we can prove that the sign-pattern of the $i$-th row of Jacobian matrix, denoted as a ternary vector $\boldsymbol{s}_i \in \{-, 0, +\}^N$, is orthogonal to $(\boldsymbol{x}^I - \boldsymbol{x}^K)$ (see Eq. (S3) in Supplementary Note 1). In other words, we can always find a real-valued vector $\boldsymbol{y} \in \mathbb{R}^N$, which has the same sign-pattern as $\boldsymbol{s}_i$ and satisfies $\boldsymbol{y}^\top \cdot (\boldsymbol{x}^I - \boldsymbol{x}^K) = 0$. If we compute the sign-patterns of all vectors orthogonal to $(\boldsymbol{x}^I - \boldsymbol{x}^K)$ for all $I, K \in \mathcal{I}_i$, then $\boldsymbol{s}_i$ must belong to the intersections of those sign-patterns, denoted as $\hat{\mathcal{S}}_i$. In fact, as long as the number $\Omega$ of steady-state samples in $\mathcal{X}$ is above certain threshold $\Omega^*$, then $\hat{\mathcal{S}}_i$ will contain only three sign-patterns $\{-\boldsymbol{a}, \boldsymbol{0}, \boldsymbol{a}\}$ (see Remark 5 in the subsection 'Inferring the sign-pattern' of Supplementary Note 1). To decide which of these three remaining sign-patterns is the true one, we just need to know the sign of only one non-zero interaction. If such prior knowledge is unavailable, one can at least make a reasonable assumption that $s_{ii} = $ '$-$', i.e., the intra-taxa interaction $J_{ii}$ is negative (which is often required for community stability). When $\hat{\mathcal{S}}_i$ has more than three sign-patterns, we proved that the steady-state data is not informative enough in the sense that all sign-patterns in $\hat{\mathcal{S}}_i$ are consistent with the data available in $\mathcal{X}$ (see Remark 5 in the subsection 'Inferring the sign-pattern' of Supplementary Note 1). This situation is not a limitation of any inference algorithm, but of the data itself. To uniquely determine the sign-pattern in such a situation, one has to either collect more samples (thus increasing the informativeness of $\mathcal{X}$) or use a priori knowledge of non-zero interactions.

We illustrate the application of the above described method to small microbial communities with unspecified population dynamics (Fig. 1). For the two taxa community (Fig. 1a), there are three possible types of equilibria, i.e., $\{\boldsymbol{x}^{\{1\}}, \boldsymbol{x}^{\{2\}}, \boldsymbol{x}^{\{1,2\}}\}$, depicted as colored pie charts in Fig. 1b. In order to infer $\boldsymbol{s}_1 = (\text{sign}(J_{11}), \text{sign}(J_{12}))$, we compute a straight line (shown in green in Fig. 1b) that is orthogonal to the vector $(\boldsymbol{x}^{\{1,2\}} - \boldsymbol{x}^{\{1\}})$ and passes through the origin. The regions (including the origin and two quadrants) crossed by this green line provide the set of possible sign-patterns $\hat{\mathcal{S}}_1 = \{(-, +), (0, 0), (+, -)\}$ that $\boldsymbol{s}_1$ may belong to. A priori knowing that $J_{11} < 0$, our method correctly concludes that $\boldsymbol{s}_1 = (-, +)$. Note that $J_{12} > 0$ is consistent with the observation that with the presence of taxon 2, the steady-state abundance of taxon 1 increases (Fig. 1b), i.e., taxon 2 promotes the growth of taxon 1. We can apply the same method to infer the sign-pattern of $\boldsymbol{s}_2 = (-, -)$.

For the three taxa community (Fig. 1c), there are seven possible types of equilibria (steady-state samples), i.e., $\{\boldsymbol{x}^{\{1\}}, \boldsymbol{x}^{\{2\}}, \boldsymbol{x}^{\{3\}}, \boldsymbol{x}^{\{1,2\}}, \boldsymbol{x}^{\{1,3\}}, \boldsymbol{x}^{\{2,3\}}, \boldsymbol{x}^{\{1,2,3\}}\}$. Four of them share taxon 1 (see colored pie charts in Fig. 1d). Six line segments connect the $\binom{4}{2} = 6$ sample pairs, and represent vectors of the form $(\boldsymbol{x}^I - \boldsymbol{x}^K), I, K \in \mathcal{I}_1 = \{\{1\}, \{1,2\}, \{1,3\}, \{1,2,3\}\}$. Considering the line segment $(\boldsymbol{x}^{\{1,3\}} - \boldsymbol{x}^{\{1\}})$, i.e., the solid blue line in Fig. 1d, we compute a plane (shown in orange in Fig. 1d) that is orthogonal to it and passes through the origin. The regions (including the origin and eight orthants) crossed by this orange plane provide a set of possible sign-patterns that $\boldsymbol{s}_1$ may belong to (Fig. 1d). We repeat the same procedure for all other vectors $(\boldsymbol{x}^I - \boldsymbol{x}^K), I, K \in \mathcal{I}_1$, and compute the intersection of all the possible sign-patterns, finally yielding the minimum set $\hat{\mathcal{S}}_1 = \{(-, 0, +), (0, 0, 0), (+, 0, -)\}$ to which $\boldsymbol{s}_1$ may belong to. If the sign of one non-zero interaction is known ($J_{11} < 0$ for this example), our method correctly infers the true sign-pattern $\boldsymbol{s}_1 = (-, 0, +)$. Repeating this process for samples sharing taxon 2 (or 3) will enable us to infer the sign-pattern $\boldsymbol{s}_2$ (or $\boldsymbol{s}_3$), respectively.

It is straightforward to generalize the above method to a microbial community of $N$ taxa (see Methods and the subsection 'Brute-force algorithm' of Supplementary Note 2 for details). But this brute-force method requires us to calculate all the sign-pattern candidates first, and then calculate their intersection to determine the minimum set $\hat{\mathcal{S}}_i$ that $\boldsymbol{s}_i$ will belong to. Since the solution space of sign-patterns is of size $3^N$, the time complexity of this brute-force method is exponential with $N$, making it impractical for a microbial community with $N > 10$ taxa (Methods). To resolve this issue, we developed a heuristic algorithm that pre-calculates many intersection lines of $(N - 1)$ non-parallel hyperplanes that pass through the origin and are orthogonal to $(\boldsymbol{x}^I - \boldsymbol{x}^K), I, K \in \mathcal{I}_i$. Based on these pre-calculated intersection lines, the algorithm determines $\hat{\mathcal{S}}_i$ using the most probable intersection line. The solution space of this heuristic algorithm is determined by the user-defined number of pre-calculated interaction lines (denoted as $\Psi$). Hence this algorithm naturally avoids searching the exponentially large solution space (see Methods and the subsection 'Inference using the heuristic algorithm' of Supplementary Note 2 for details). Later on, we will show that this heuristic algorithm can indeed infer the interaction types with high accuracy.

In reality, due to measurement noise and/or transient behavior of the microbial community, the abundance profiles of the collected samples may not exactly represent steady states of the microbial community. Hence, for certain $J_{ij}$'s, their inferred signs might be wrong. Using simulated data, we will show later that for considerable noise level the inference accuracy is still reasonably high.

**Inferring interaction strengths**. To quantitatively infer the inter-taxa interaction strengths, it is necessary to choose a priori a parameterized dynamic model for the microbial community. The

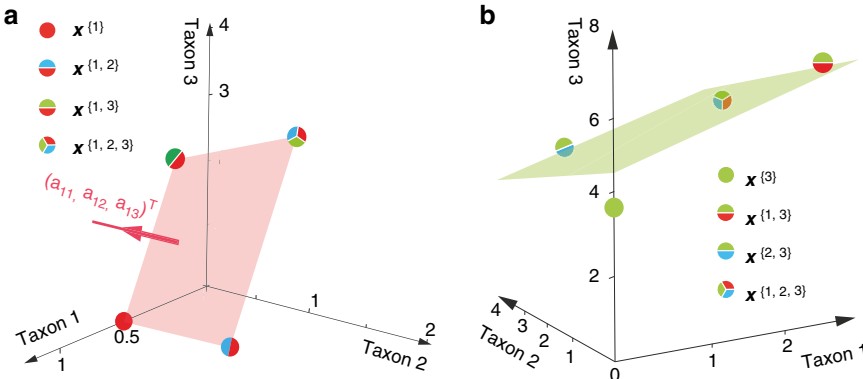

**Fig. 2** Consistency check of the GLV model and the observed steady-state samples. For a microbial community following exactly the GLV dynamics, all its steady-state samples sharing one common taxon will align onto a hyperplane in the state space. **a** Here we consider a microbial community of three taxa. There are four steady-state samples $\{x^{\{1\}}, x^{\{1,2\}}, x^{\{1,3\}}, x^{\{1,2,3\}}\}$ that share common taxon 1. Those four steady-state samples represent four points in the state space, and they align onto a plane (light red). The normal vector of this plane is parallel to the first row $a_1$ of the interaction matrix $A$ in the GLV model. Given any one of non-zero entries in $a_1$, we can determine the exact values of all other entries. Otherwise, we can always express the inter-taxa interaction strengths $a_{ij}$ ($j \neq i$) as a function of the intra-taxa interaction strength $a_{ii}$. **b** Here we again consider a microbial community of three taxa. Taxon-1 and taxon-2 follow the GLV dynamics, but taxon-3 does not. Then those steady-state samples that share taxon-3 do not align onto a plane anymore. Here we show the best fitted plane (in green) by minimizing the distance between this plane and the four steady states, with the coefficient of determination $R^2 = 0.77$

classical GLV model can be obtained from Eq. (1) by choosing

$$f_i(x) = \sum_{j=1}^{N} a_{ij}x_j + r_i, \quad i = 1, \ldots, N, \quad (2)$$

where $r = (r_1, \ldots, r_N)^T \in \mathbb{R}^N$ is the intrinsic growth rate vector and $A = (a_{ij}) \in \mathbb{R}^{N \times N}$ is the interaction matrix characterizing the intra- and inter-taxa interactions.

From Eq. (2) we can easily calculate the Jacobian matrix $J$, which is nothing but the interaction matrix $A$ itself. This also reflects the fact that the value of $a_{ij}$ quantifies the interaction strength of taxon $j$ on taxon $i$. The GLV model considerably simplifies the inference of the ecological network, because we can prove that $a_i \cdot (x^I - x^K) = 0$, for all $I, K \in \mathcal{I}_i$, where $a_i \equiv (a_{i1}, \ldots, a_{iN})$ represents the $i$-th row of the $A$ matrix (see the subsection 'Inference of interaction strengths and intrinsic growth rates' of Supplementary Note 5). In other words, all steady-state samples containing the $i$-th taxon will align exactly onto a hyperplane, whose orthogonal vector is parallel to the vector $a_i$ that we aim to infer (Fig. 2a, Theorem 3 of the subsection 'A condition for detecting GLV dynamics' of Supplementary Note 5). Thus, for the GLV model, the inference from steady-state data reduces to finding an $(N - 1)$-dimensional hyperplane that 'best fits' the steady-state sample points $\{x^I | I \in \mathcal{I}_i\}$ in the $N$-dimensional state space. In order to exactly infer $a_i$, it is necessary to know the value of at least one non-zero element in $a_i$, say, $a_{ii}$. Otherwise, we can just determine the relative interaction strengths by expressing $a_{ij}$ in terms of $a_{ii}$. Once we obtain $a_i$, the intrinsic growth rate $r_i$ of the $i$-th taxon can be calculated by averaging $(-a_i \cdot x^I)$ overall $I \in \mathcal{I}_i$, i.e., all the steady-state samples containing taxon $i$.

In case the samples are not collected exactly at steady states of the microbial community or there is noise in abundance measurements, those samples containing taxon $i$ will not exactly align onto a hyperplane. A naive solution is to find a hyperplane that minimizes its distance to those noisy samples. But this solution is prone to induce false-positive errors and will yield non-sparse solutions (corresponding to very dense ecological networks). This issue can be partly alleviated by introducing a Lasso regularization[40], implicitly assuming that the interaction matrix $A$ in the GLV model is sparse. However, the classical Lasso regularization may induce a high false-discovery rate (FDR), meaning that many zero interactions are inferred as non-zeros ones. To overcome this

drawback, we applied the Knockoff filter[41] procedure, allowing us to control the FDR below a desired user-defined level $q > 0$ (see the subsection 'Applying the Knockoff filter to control the false-discovery rate' of Supplementary Note 5 for details).

The observation that for the GLV model all noiseless steady-state samples containing the $i$-th taxon align exactly onto a hyperplane can also be used to characterize how much the dynamics of the $i$-th taxon in a real microbial community deviates from the GLV model. This deviation can be quantified by the coefficient of determination (denoted by $R^2$) of the multiple linear regression when fitting the hyperplane using the steady-state samples (Fig. 2b). If $R^2$ is close to 1 (the samples indeed align to a hyperplane), we conclude that the dynamics of the microbial community is consistent with the GLV model, and hence the inferred interaction strengths and intrinsic growth rates are reasonable. Otherwise, we should only aim to qualitatively infer the ecological interaction types that do not require specifying any population dynamics.

**Validation on simulated data**. To validate the efficacy of our method in inferring ecological interaction types, we numerically calculated the steady states of a small microbial community with $N = 8$ taxa, using four different population dynamics models[35–39]: GLV, Holling Type II (Holling II), DeAngelis–Beddington (DB), and Crowley–Martin (CM) models (see Supplementary Note 4 for details). Note that all these models satisfy the requirement that the sign-pattern of the Jacobian matrix is time-invariant. To infer the ecological interaction types among the eight taxa, we employed both the brute-force algorithm (with solution space $\sim 3^8 = 6,561$) and the heuristic algorithm (with solution space given by the number of the pre-calculated intersections chosen as $\Psi = 5N = 40$).

In the noiseless case, we find that when the number of steady-state samples satisfies $\Omega > 3N$, the heuristic algorithm outperformed the brute-force algorithm for data sets generated from all the four different population dynamics models (Fig. 3a). This result is partly due to the fact that the former requires many fewer samples than the latter to reach high accuracy (the percentage of correctly inferred interaction types). However, when the sample size $\Omega$ is small ($<3N$), the heuristic algorithm completely fails while the brute-force algorithm still works to some extent.

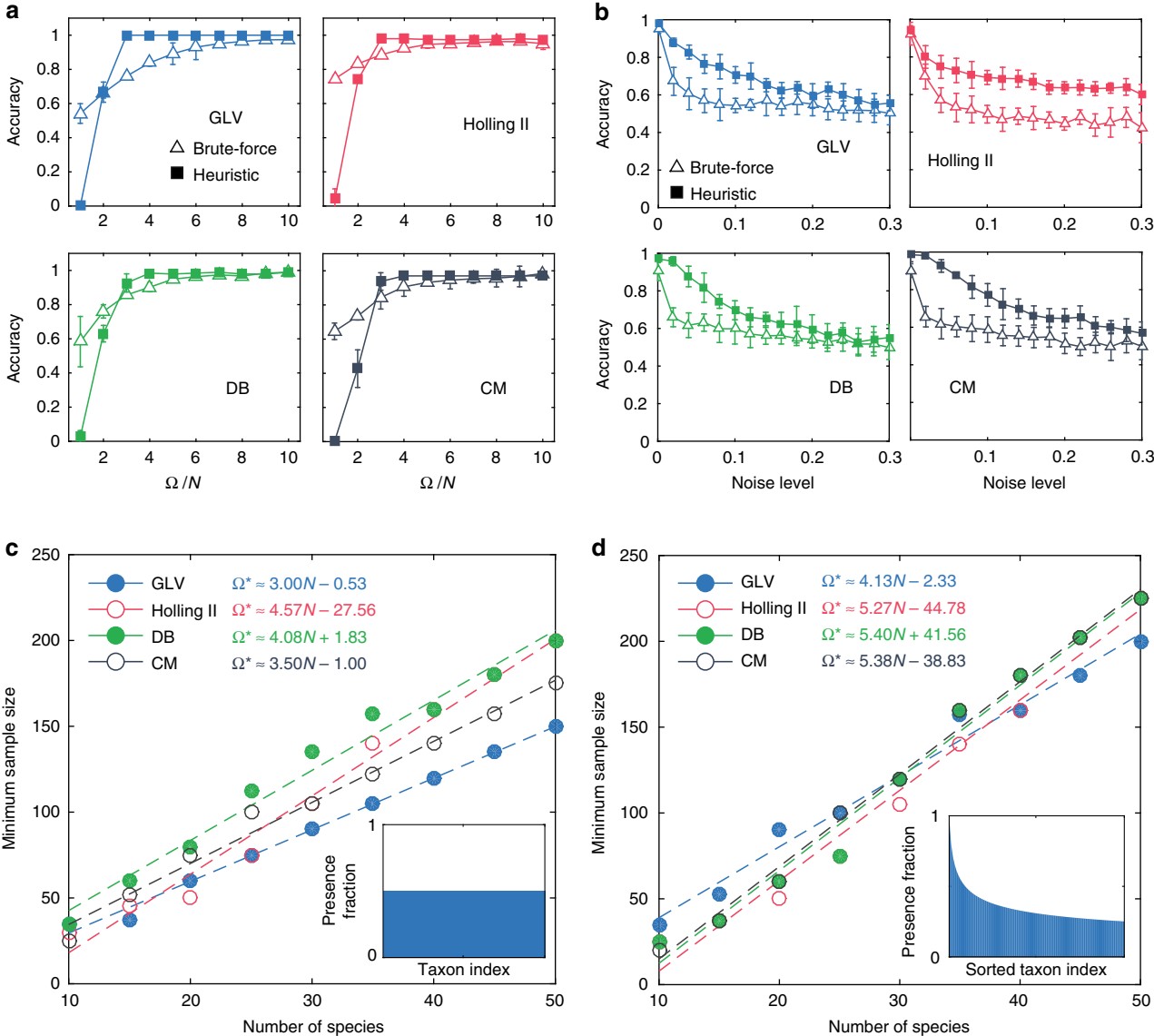

**Fig. 3** Validation of our method in inferring interaction types using simulated data. **a,b** Consider a small microbial community of $N = 8$ taxa. We generate steady-state samples using four different population dynamics models: Generalized Lotka–Volterra (GLV), Holling Type II (Holling II), DeAngelis–Beddington (DB), and Crowley–Martin (CM). We compare the performance of the brute-force algorithm (with solution space $\sim 3^8 = 6,561$) and the heuristic algorithm (with solution space $\sim \Psi = 5N = 40$). **a** In the noiseless case, we plot the inference accuracy (defined as the percentage of correctly inferred signs in the Jacobian matrix) as a function of sample size $\Omega$. **b** In the presence of noise, we plot the inference accuracy as a function of the noise level $\eta$. Here the sample size is fixed: $\Omega = 5N = 40$. The error bar represents standard deviation for 10 different realizations. **c, d** We calculate the minimal sample size $\Omega^*$ required for the heuristic algorithm to achieve high accuracy (100% for GLV, 95% for Holling II, DB, and CM) at different system sizes. We consider two different taxa presence patterns: uniform and heterogenous (see insets). Here the simulated data is generated in the noiseless case and we chose $\Psi = 10N$. **a–d** The underlying ecological network is generated from a directed random graph model with connectivity 0.4 (i.e., with probability 0.4 there will be a directed edge between any two taxa)

We then fix $\Omega = 5N$, and compare the performance of the brute-force and heuristic algorithms in the presence of measurement noise (Fig. 3b). We add artificial noise to each non-zero entry $x_i^I$ of a steady-state sample $\boldsymbol{x}^I$ by replacing $x_i^I$ with $x_i^I + \eta u$, where $u \sim U[-x_i^I, x_i^I]$ is a random number uniformly distributed in the interval $[-x_i^I, x_i^I]$ and $\eta \geq 0$ quantifies the noise level. We again find that the heuristic algorithm works better than the brute-force algorithm for data sets generated from all the four different population dynamics models.

The above encouraging results on the heuristic algorithm prompt us to systematically study the key factor to obtain an accurate inference, i.e., the minimal sample size $\Omega^*$ (Fig. 3c, d). Note that for a microbial community of $N$ taxa, if we assume that for any subset of the $N$ taxa there is only one stable steady state

such that all the corresponding taxa have non-zero abundance, then there are at most $\Omega_{max} = (2^N - 1)$ possible steady-state samples. (Of course, not all of them will be ecologically feasible. For example, certain pairs of taxa will never coexist.) In general, it is unnecessary to collect all possible steady-state samples to obtain a highly accurate inference result. Instead, we can rely on a subset of them. To demonstrate this, we numerically calculated the minimal sample size $\Omega^*$ we need to achieve a highly accurate inference of interaction types. We considered two different taxa presence patterns: (1) uniform: all taxa have equal probability of being present in the steady-state samples (inset of Figs. 3c); and (2) heterogenous: a few taxa have higher presence probability than others, reminiscent of human gut microbiome samples[26] (inset of Fig. 3d). We found that for the steady-state data

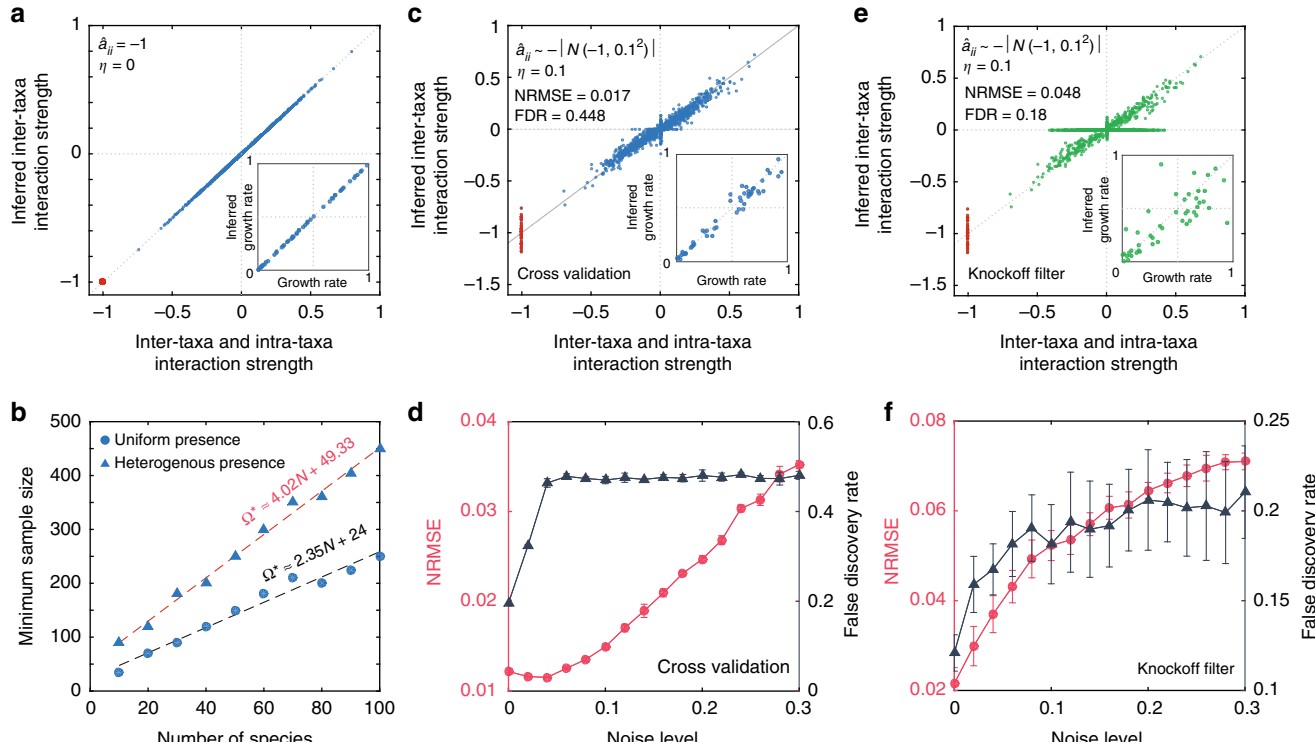

**Fig. 4** Validation of our method in inferring interaction strengths using simulated data. Here, we simulate steady-state samples using the GLV model with $N$ taxa and intra-taxa interaction strength $a_{ii} = -1$ for each taxon. The underlying ecological network is generated from a directed random graph model with connectivity 0.4. The noise is added to steady-state samples as follows: $x_i^l \rightarrow x_i^l + \eta u$, where the random number $u$ follows a uniform distribution $U[-x_i^l, x_i^l]$, and $\eta$ is the noise level. **a, b** Consider the ideal case: (1) noiseless $\eta = 0$; and (2) we know exactly $a_{ii} = -1$. **a** We can perfectly infer $a_{ij}$'s and $r_i$'s. **b** The minimal sample size $\Omega^*$ required to correctly infer the interaction strengths scales linearly with the system size $N$. Here we consider two different taxa presence pattern: uniform and heterogenous. **c-f** In the presence of noise, and during the we just assume that the intra-taxa interaction strengths $\hat{a}_{ii}$ follows a half-normal distribution. **c** Using the Lasso regularization induces high false-discovery rate (FDR) ~ 0.448 at $\eta = 0.1$. **d** Using classical Lasso with cross validation, both NRMSE and FDR increase with increasing $\eta$. **e** For the same data set used in **c**, we use the knockoff filter to control the FDR below a certain level $q = 0.2$. **f** With increasing noise level $\eta$, FDR can still be successfully controlled below $q = 0.2$ by applying the knockoff filter. In subfigures **a, c-f**, we have $N = 50$. The error bar represents standard deviation for 10 different realizations

generated from all the four population dynamics models, $\Omega^*$ always scales linearly with $N$ in both taxa presence patterns, and the uniform taxa presence pattern requires much fewer samples (Fig. 3c, d).

Note that as $N$ grows, the total possible steady-state samples $\Omega_{max}$ increases exponentially, while the minimal sample size $\Omega^*$ we need for high inference accuracy increase linearly. Hence, interestingly, we have $\Omega^*/\Omega_{max} \rightarrow 0$ as $N$ increases. This suggests that as the number of taxa increases, the proportion of samples needed for accurate inference actually decreases. This is a rather counter-intuitive result because, instead of a 'curse of dimensionality', it suggests that a 'blessing of dimensionality' exists when using the heuristic algorithm to infer interaction types for microbial communities with a large number of taxa.

To validate our method in quantitatively inferring inter-taxa interaction strengths, we numerically calculated steady states for a microbial community of $N = 50$ taxa, using the GLV model with $a_{ii} = -1$ for all taxa.

In the noiseless case, if during the inference we know exactly $a_{ii} = -1$ for all taxa, then we can perfectly infer the inter-taxa interaction strengths $a_{ij}$'s and the intrinsic growth rates $r_i$'s (see Fig. 4a). To study the minimal sample size $\Omega^*$ required for perfect inference in the noiseless case, we again consider two different taxa presence patterns: (1) uniform and (2) heterogeneous. We find that for both taxa presence patterns $\Omega^*$ scales linearly with $N$, though the uniform taxa presence pattern requires much fewer samples (Fig. 4b).

In the presence of noise, and if we don't know the exact values of $a_{ii}$'s, but just assume they follow a half-normal distribution $-|\mathcal{N}(-1, 0.1^2)|$, we can still infer $a_{ij}$'s and $r_i$'s with reasonable accuracy (with the normalized root-mean-square error (NRMSE) $< 0.08$), for noise level $\eta < 0.3$ (Fig. 4c-f). However, we point out that the classical Lasso regularization could induce many false positives, and the FDR reaches 0.448 at noise level $\eta = 0.1$, indicating that almost half of inferred non-zero interactions are actually zero (Fig. 4c). Indeed, even with a noise level $\eta = 0.04$, the classical Lasso already yields FDR $\sim 0.45$, staying there for higher $\eta$ (Fig. 4d).

In many cases, we are more concerned about low FDR than high false-negative rates, because the topology of an inferred ecological network with even many missing links can still be very useful in the study of its dynamical and control properties[42]. To control FDR below a certain desired level $q = 0.2$, we applied the Knockoff filter[41] (Fig. 4e), finding that though it will introduce more false negatives (see the horizontal bar in Fig. 4e), it can control the FDR below 0.2 for a wide range of noise level (Fig. 4f).

We also found that applying this GLV inference method to samples obtained from a microbial community with non-GLV dynamics leads to significant inference errors even in the absence of noise (Supplementary Fig. 9).

**Application to experimental data.** First, we applied our inference method to an experimental data set from a synthetic soil microbial

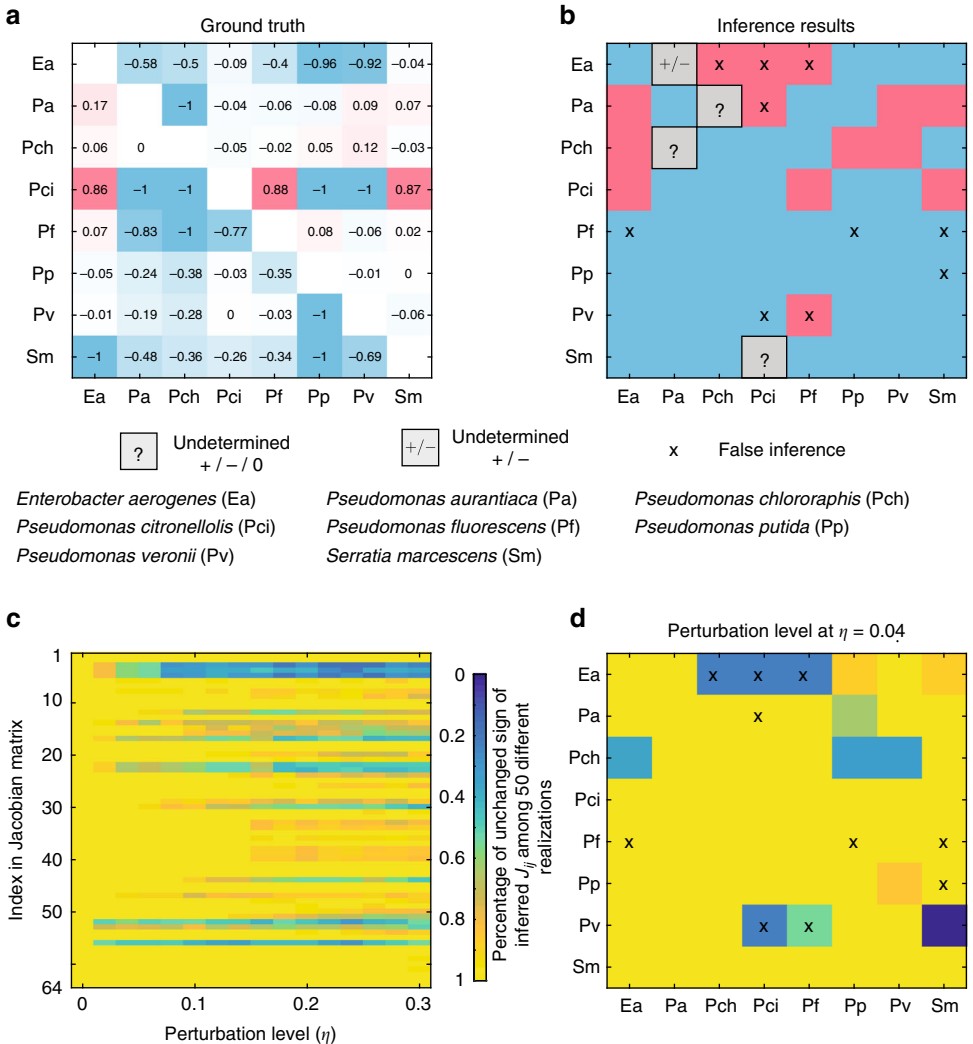

**Fig. 5** Inferring interaction types of a synthetic soil microbial community. The steady-state samples were experimentally collected from a synthetic soil microbial community of eight bacterial species. Those steady-state samples involve 101 different species combinations: all eight solos, 28 duos, 56 trios, all eight septets, and 1 octet. **a** From the eight solos (monoculture experiments) and 28 duos (pair-wise co-culture experiments), one can calculate the relative yield $R_{ij}$, quantifying the promotion (positive) or inhibition (negative) impact of species $j$ on species $i$. The values shown in the relative yield matrix $R = (R_{ij})$ quantify the strengths of promotion and inhibition effects. The sign-pattern of this matrix serves as the ground truth of that of the Jacobian matrix associated with the unknown population dynamics of this microbial community. **b** Without considering the 8 solos and 28 duos, we analyze the other steady-state samples. We use the brute-force method to infer the ecological interaction types, i.e., the sign-pattern of the Jacobian matrix. Blue (or red) means inhibition (or promotion) effect of species $j$ on species $i$, respectively. 10 signs (labeled by '×') are falsely inferred, four signs (gray) are undetermined by the analyzed steady-state samples. **c, d** The robustness of the inference results in the presence of artificially added noise: $x_i^l \rightarrow x_i^l + \eta u$, where the random number $u$ follows a uniform distribution $U[-x_i^l, x_i^l]$, and $\eta$ is the noise level. At each noise level, we run 50 different realizations. Yellow (or blue) means many (or few) inferred $J_{ij}$ keep the same sign among 50 different realizations. **c** Many of the inferred $J_{ij}$ keep their signs in the presence of noise up to noise level $\eta = 0.3$. **d** At $\eta = 0.04$, we plot the percentage of unchanged signs for inferred Jacobian matrix in 50 different realizations. The '×' labels correspond to the 10 falsely inferred signs shown in **b**. Five of the 10 falsely inferred interactions change their signs frequently even when the perturbation is very small, implying that the falsely inferred signs in **b** could be due to measurement noise in the experiments

community of eight bacterial species[43]. This data set consists of steady states of a total of 101 different species combinations: all eight solos, 28 duos, 56 trios, all eight septets, and one octet (see the subsection 'A synthetic microbial community of 8 soil bacteria' of Supplementary Note 6 for details). For those steady-state samples that started from the same species collection, but with different initial conditions, we average over their final steady states to get a representative steady state for this particular species combination.

In the experiment, it was found that several species grew to a higher density in the presence of an additional species than in monoculture. The impact of each additional species (competitor) $j$ on each focal species $i$ can be quantified by the so-called relative yield, defined as: $R_{ij} = \frac{x_i^{\{i,j\}} - x_i^{\{i\}}}{x_i^{\{i,j\}} + x_i^{\{i\}}}$, which represents a proxy of the ground truth of the interaction strength that species $j$ impacts species $i$. A negative relative yield indicates growth hindrance of species $j$ on $i$, whereas positive values indicated facilitation (Fig. 5a). Though quantifying the relative yield is conceptually easy and implementable for certain small microbial communities (see Supplementary Note 7 for details), for many host-associated microbial communities with many taxa, such as the human gut microbiota, measuring these one-species and two-species samples is simply impossible. This actually motivates the inference method we developed here.

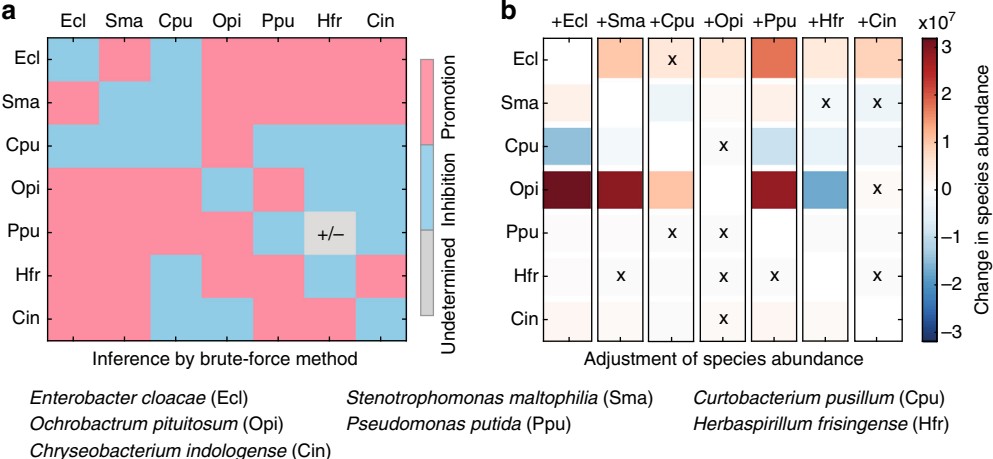

**Fig. 6** Inferring interaction types in a synthetic community of maize roots with seven bacterial species. The data set consists of seven sextets and one septet. **a** Without considering the one septet, we analyze the seven sextets (steady-state samples involving six of the seven species). We use the brute-force method to infer the ecological interaction types, i.e., the sign-pattern of the Jacobian matrix. Blue (or red) means inhibition (or promotion) effect of species $j$ on species $i$, respectively. One sign (gray) is undetermined from the seven sextets. **b** The changes of species abundance before and after, respectively, adding one species into a sextet. Blue (or red) corresponds to the decrease (or increase) of one species' abundance after introducing a new species into sextets, respectively. Each column corresponds to a sextet (a six-baterial community), the name of the newly introduced species is marked in the top of each column. '×' indicates the false prediction. There are in total 12 false predictions

Before we apply our inference method, to be fair, we remove all those steady states involving one or two species, and analyze only the remaining 65 steady states. (Note that for $N = 8$, the number of total possible steady states is $\Omega_{max} = 255$. Hence we only use roughly one quarter of the total possible steady states.) During the inference, we first check if the population dynamics of this microbial community can be well described by the GLV model. We find that all the fitted hyperplanes show small $R^2$, indicating that the GLV model is not appropriate to describe the dynamics of this microbial community (Supplementary Fig. 10b). Hence, we have to aim for inferring the ecological interaction types, without assuming any specific population dynamics model.

Since this microbial community has only eight species, we can use the brute-force algorithm to infer the sign-pattern of the $8 \times 8$ Jacobian matrix, i.e., the ecological interaction types between the eight species (The results of using a heuristic algorithm are similar and described in the Supplementary Fig. 10c). Compared with the ground truth obtained from the relative yield matrix (Fig. 5a), we find that 50 (78.13%) of the 64 signs were correctly inferred, 10 (15.62%) signs were falsely inferred (denoted as '×'), and four (6.25%) signs cannot be determined (denoted as '?') with the information provided by the 65 steady states (Fig. 5b).

We notice that the relative yield of many falsely inferred interactions is weak (with the exception of $R_{Ea,Pch}$, and $R_{Ea,Pf}$). We conjecture that these errors are caused by noise or measurement errors in the experiments. To test this conjecture, we analyzed the robustness of each inferred $s_{ij}$ by calculating the percentage of unchanged $s_{ij}$ after adding perturbations (noise) to the samples (Fig. 5c). Similar to adding noise to simulated steady-state data, here we add noise to each non-zero entry $x_i^I$ of a sample $\boldsymbol{x}^I$ such that $x_i^I \rightarrow x_i^I + \eta u$, where $u \sim U[-x_i^I, x_i^I]$. The more robust the inferred results are, the higher the percentage of unchanged signs as $\eta$ is increased. We found that most of the inferred signs were robust: the percentage of unchanged signs remained nearly 80% up to noise level $\eta = 0.3$ (Fig. 5c). Specifically, Fig. 5d plots the percentage of unchanged signs of the inferred Jacobian matrix when $\eta = 0.04$. We found that even if the perturbation is very small, 5 of the 10 falsely inferred $s_{ij}$ in Fig. 5b changed their signs very frequently (blue entries with label '×' in Fig. 5d). In other

words, those interactions were very sensitive to noise, suggesting that some falsely inferred signs in Fig. 5b were largely caused by measurement noise.

Second, we applied our method to an experimental data set from a synthetic bacterial community of maize roots[44]. There are seven bacterial species (Ecl, Sma, Cpu, Opi, Ppu, Hfr, and Cin) in this community. This data set consists of in total eight steady-state samples: seven sextets and one septet. We found that this community cannot be described by the GLV dynamics (Supplementary Fig. 11).

Using only the seven sextets (i.e., seven steady-state samples involving six of the seven species), we inferred the sign-pattern of the Jacobian matrix (Fig. 6a). Based on the sign of $J_{ij}$, we can predict how the abundance of species-$i$ in a microbial community will change, when we add species-$j$ to the community. For example, if we add Ecl to a community consisting of the other six species (i.e., Sma, Cpu, Opi, Ppu, Hfr, and Cin), we predict that the abundance of Sma, Opi, Ppu, Hfr, and Cin will increase, while the abundance of Cpu will decrease (first column of Fig. 6b). Note that our prediction only considers the direct ecological interactions between species and ignores the indirect impact among species. Indeed, Ecl promotes Opi, but Ecl also promotes Hfr that inhibits Opi. Hence the net effect of Ecl on Opi is hard to tell without knowing the interaction strengths. Nevertheless, we found that our prediction is consistent with experimental observation (Fig. 6b, first column).

We then systematically compared our predictions of species abundance changes with experimental observations. There are in total seven sextets, corresponding to the seven columns in Fig. 6b. We add the corresponding missing species back to the community, and check the abundance changes of the existing six species. There are in total $6 \times 7 = 42$ abundance changes. We found that our inferred sign-pattern of the Jacobian matrix (Fig. 6a) can correctly predict 30 of the 42 abundance changes (accuracy ~71.43%). Moreover, for those false predictions, the detailed values of the abundance changes are actually relatively small (comparing to those of correct predictions). Note that we only used seven steady-samples to infer the interaction types. If we have more steady-state samples available, we assume the prediction accuracy of our method can be further improved.

We also demonstrated the application of our inference method to two additional experimental data sets. One was obtained from a synthetic microbial community of two cross-feeding partners[34], the other was obtained from a synthetic community of 14 auxotrophic *Escherichia coli* strains[45]. For both data sets, our inference results agree well with the ground truth (see Supplementary Note 6 for details).

## Discussion

In this work, we developed a new inference method to map the ecological networks of microbial communities using steady-state data. Our method can qualitatively infer ecological interaction types (signs) without specifying any population dynamics model. Furthermore, we show that steady-state data can be used to test if the dynamics of a microbial community can be well described by the classic GLV model. When GLV is found to be adequate, our method can quantitatively infer inter-taxa interaction strengths and the intrinsic growth rates.

The proposed method bears some resemblance to previous network reconstruction methods based on steady-state data[46]. But we emphasize that, unlike the previous methods, our method does not require any perturbations applied to the system nor sufficiently close steady states. For certain microbial communities such as the human gut microbiota, applying perturbations may raise severe ethical and logistical concerns.

Note that our method requires the measurement of steady-state samples and absolute taxon abundances. For systems that are in frequent flux, where steady-state samples are hard to collect, our method is not applicable. Moreover, it fails on analyzing the relative abundance data (see Methods). Note that the compositionality of relative abundance profiles also represents a major challenge for inference methods based on temporal data[15,19]. Fortunately, for certain small laboratory-based microbial communities, we can measure the absolute taxon abundances in a variety of ways, e.g., selective plating[47], quantitative polymerase chain reaction (qPCR)[15,16,48,49], flow cytometry[50], and fluorescence in situ hybridization (FISH)[51]. For example, in the study of a synthetic soil microbial community of eight bacterial species[43], the total cell density was assessed by measuring the optical density and species fractions (relative abundance) were determined by plating on nutrient agar plates. In recent experiments evaluating the dynamics of *Clostridium difficile* infection in mice models[15,16], two sources of information were combined to measure absolute abundances: (1) data measuring relative abundances of microbes, typically consisting of counts (e.g., high-throughput 16S rRNA sequencing data); and (2) data measuring overall microbial biomass in the sample (e.g., universal 16S rRNA qPCR).

In contrast to the difficulties encountered in attempts to enhance the informativeness of temporal data that are often used to infer ecological networks of microbial communities, the informativeness of independent steady-state data can be enhanced by simply collecting more steady-state samples with distinct taxa collection. For host-associated microbial communities, this can be achieved by collecting steady-state samples from different hosts. Our numerical analysis suggests that the minimal number of samples with distinct taxa collections required for robust inference scales linearly with the taxon richness of the microbial community. Our analysis of experimental data from a small synthetic microbial community of eight species shows that collecting roughly one quarter of the total possible samples is enough to obtain a reasonably accurate inference. Furthermore, our numerical results suggest that this proportion can be significantly lower for larger microbial communities.

This blessing of dimensionality suggests that our method holds great promise for inferring the ecological networks of large and complex microbial communities, such as the human gut microbiota. There are two more encouraging facts that support this idea. First of all, it has been shown that the composition of the human gut microbiome remains stable for months and possibly even years until a major perturbation occurs through either antibiotic administration or drastic dietary changes[52–55]. The striking stability and resilience of human gut microbiota suggest that the collected samples very likely represent the steady states of the gut microbial ecosystem. Second, for healthy adults the gut microbiota displays remarkable universal ecological dynamics[56] across different individuals. This universality of ecological dynamics suggests that microbial abundance profiles of steady-state samples collected from different healthy individuals can be roughly considered as steady states of a conserved 'universal gut dynamical' ecosystem and hence can be used to infer its underlying ecological network. Despite the encouraging facts, we emphasize that there are still many challenges in applying our method to infer the ecological network of the human gut microbiota. For example, the assumption of invariant ecological interaction types (i.e., promotion, inhibition, or neutral) between any two taxa needs to be carefully verified. Moreover, our method requires the measurement of absolute abundances of taxa.

We expect that additional insights into microbial ecosystems will emerge from a comprehensive understanding of their ecological networks. Indeed, inferring ecological networks using the method developed here will enable enhanced investigation of the stability[57] and assembly rules[58] of microbial communities as well as facilitate the design of personalized microbe-based cocktails to treat diseases related to microbial dysbiosis[9,10].

## Methods

**Theoretical basis for inferring ecological interactions**. Consider a microbial community of $N$ taxa, whose population dynamics follow Eq.(1). A steady-state data set $\mathcal{X}$ is a collection of $N$-dimensional vectors $\boldsymbol{x} \in \mathbb{R}^N$ corresponding to the measured equilibria of Eq. (1). We will denote a steady-state sample as $\boldsymbol{x}^I \in \mathbb{R}^N$, where the superscript $I \in \mathcal{I}$ determines which taxa are present, and $\mathcal{I} = 2^{\{1,\ldots,N\}}$ is the set of all possible subsets of $\{1, \ldots, N\}$. Consider the subset $\mathcal{X}_i \subseteq \mathcal{X}$ of all samples containing taxon $i$ so that $f_i(\boldsymbol{x}) = 0$ for all $\boldsymbol{x} \in \mathcal{X}_i$. Applying the mean value theorem for multivariable functions, we obtain

$$f_i(\boldsymbol{x}^I) - f_i(\boldsymbol{x}^K) = \left( \int_0^1 \frac{\partial f_i(\boldsymbol{x}^I + \sigma(\boldsymbol{x}^K - \boldsymbol{x}^I))}{\partial \boldsymbol{x}} \, \mathrm{d}\sigma \right) \cdot \left( \boldsymbol{x}^I - \boldsymbol{x}^K \right) = 0, \quad (3)$$

where '·' denotes the inner product between vectors in $\mathbb{R}^N$. Let $\boldsymbol{J}_i(\boldsymbol{x}) = \frac{\partial f_i(\boldsymbol{x})}{\partial \boldsymbol{x}} \in \mathbb{R}^N$ be the $i$-th row of the Jacobian matrix $J(\boldsymbol{x}) = \left(J_{ij}(\boldsymbol{x})\right) = \left(\frac{\partial f_i(\boldsymbol{x})}{\partial x_j}\right)$ and let us introduce the notation

$$\int_{L_{\boldsymbol{x}^I, \boldsymbol{x}^K}} \boldsymbol{J}_i := \int_0^1 \boldsymbol{J}_i(\boldsymbol{x}^I + \sigma(\boldsymbol{x}^K - \boldsymbol{x}^I)) \mathrm{d}\sigma, \quad (4)$$

where $L_{\boldsymbol{x}^I, \boldsymbol{x}^K}$ denotes the line segment connecting the points $\boldsymbol{x}^I$ and $\boldsymbol{x}^K$ in $\mathbb{R}^N$. With this notation, Eq. (3) can be rewritten more compactly as

$$\left( \int_{L_{\boldsymbol{x}^I, \boldsymbol{x}^K}} \boldsymbol{J}_i \right) \cdot \left( \boldsymbol{x}^I - \boldsymbol{x}^K \right) = 0, \quad \forall \boldsymbol{x}^I, \boldsymbol{x}^K \in \mathcal{X}_i. \quad (5)$$

The above equation implies that the difference of any two samples $\{\boldsymbol{x}^I, \boldsymbol{x}^K\}$ sharing taxon $i$ constrains the integral of $\boldsymbol{J}_i$ over the line segment joining them $\boldsymbol{x}^I - \boldsymbol{x}^K$.

We consider that the ecological interactions in a microbial community are encoded in the Jacobian matrix $J \in \mathbb{R}^{N \times N}$ of its population dynamics. More precisely, we assume that the $j$-th taxon directly impacts the $i$-th one iff the function $J_{ij}(\boldsymbol{x}) \not\equiv 0$. Notice that this condition is well defined because $J_{ij}(\boldsymbol{x})$ is a meromorphic function. Furthermore, an ecological interaction is inhibitory iff $J_{ij}(\boldsymbol{x}) < 0$ and excitatory iff $J_{ij}(\boldsymbol{x}) > 0$.

Inferring the absence or presence of interactions is equivalent to inferring the zero-pattern of the Jacobian matrix, recovering the topology of the ecological network underlying the microbial community. Furthermore, inferring the type of interactions (inhibitory, excitatory, or null) is equivalent to inferring the sign-pattern of the Jacobian matrix.

**Inference using the brute-force algorithm.** In Theorem 2 of the subsection 'Inferring the sign-pattern' of Supplementary Note 1, in order to check if there is a vector with sign-pattern $\boldsymbol{s} \in \{-, 0, +\}^N$ orthogonal to a given vector $(\boldsymbol{x}^I - \boldsymbol{x}^K)$, we can check if the following linear program has a solution:

$$\text{Find } \boldsymbol{v} \in \mathbb{R}^N \text{ subject to } \boldsymbol{v}^T(\boldsymbol{x}^I - \boldsymbol{x}^K) = 0 \text{ and } \text{sign}(\boldsymbol{v}) = \boldsymbol{s}. \quad (6)$$

Note that the condition $\text{sign}(\boldsymbol{v}) = \boldsymbol{s}$ can be encoded as a set of equalities/inequalities of the form $\{v_i = 0, v_i < 0, v_i > 0\}$ corresponding to the cases $\{s_i = 0, s_i = -1, s_i = 1\}$. Therefore, we can construct an algorithm to obtain all admissible sign-patterns for given steady-state data. Indeed, by enumerating all possible sign-patterns, we can use the liner program in Eq. (6) to check if each of the possible $3^N$ sign-patterns is admissible for taxon $i$. See Algorithm 1 in the subsection 'Brute-force algorithm' of Supplementary Note 2 for the pseudo code. This brute-force algorithm relies on the enumeration of all $3^N$ possible sign-patterns in $\mathbb{R}^N$, since it needs to test if each one of them is admissible for the given data. If the set $\mathcal{X}_i$ has $n_i$ elements, there will be $n_i(n_i - 1)/2$ vectors of the form $\boldsymbol{x}^I - \boldsymbol{x}^K$ with $(\boldsymbol{x}^I, \boldsymbol{x}^K) \in \mathcal{X}_i \times \mathcal{X}_i$. Then for each of those vectors and each of the possible $3^N$ sign-patterns, we will need to run the linear program Eq. (6) to check if there is an orthogonal vector with the desired sign-pattern. If we assume that the linear program can be solved with $N$ operations, then for each taxon the Brute-force algorithm requires to perform a number of operations in the order of $N 3^N n_i(n_i - 1)/2$. Hence, for 100 taxa, we will need to perform at least $5.19 \times 10^{49}$ operations for the reconstruction of each taxon—which is a number with the same order of magnitude as the number of atoms in Earth. Furthermore, the linear programming used in the brute-force method can also be time consuming even for a small microbial community with $N \sim 10$. Consequently, applying the enumeration procedure is only reasonable for a community with $N \sim 10$, since in this case only around $10^6$ operations are needed to infer the sign-pattern of the Jacobian corresponding to each taxon.

**Inference using the heuristic algorithm.** The computational complexity of the brute-force algorithm motivated us to develop a more efficient reconstruction method. This method has two main ingredients. First, a graph-based approach to quickly check whether a region can be crossed by a hyperplane, circumventing the need to solve the linear program. Second, a heuristic algorithm efficiently explores the solution space and infers the ecological interaction types.

First, we formalize a sign-satisfaction problem. Consider a real-valued vector $\boldsymbol{y} \in \mathbb{R}^N$. Thus, solving the linear program Eq. (6) is equivalent to solving the following sign-satisfaction problem:

$$\text{Find } \text{sign}(\boldsymbol{y}) \in \{-, 0, +\}^N \text{ subject to } \boldsymbol{y}^T(\boldsymbol{x}^I - \boldsymbol{x}^K) = 0. \quad (7)$$

Notice that from a geometrical viewpoint, solving Eq. (7) is equivalent to finding the orthants of $\mathbb{R}^N$ crossed by the hyperplane orthogonal to $(\boldsymbol{x}^I - \boldsymbol{x}^K)$.

Second, we propose a graph-based approach to solving the sign-satisfaction problem. For the definition, construction, and examples of sign-satisfaction graph, see the subsection 'Inference using the heuristic algorithm' of Supplementary Note 2. By using the sign-satisfaction graph, it is very efficient to test if the hyperplane orthogonal to $(\boldsymbol{x}^I - \boldsymbol{x}^K)$ crosses some orthants of $\mathbb{R}^N$, because it reduces to checking if its corresponding vector in $\{-, 0, +\}^N$ meets the requirements of sign-satisfaction. However, finding all orthants crossed by such orthogonal hyperplane remains challenging, since the sign-satisfaction graph did not decrease the dimension of the solution space (that remains with exponential size $3^N$). To address this issue, next we introduce a method to efficiently sample paths in the sign-satisfaction graph.

Third, we use the intersection line of hyperplanes to sample paths in the sign-satisfaction graph. This method depends on a user-defined parameter $\Psi \geq 1$ that specifies the number of intersection lines we need to compute. There are five steps: (step-1): Construct the matrix of the difference of all the sample pairs. Consider the set of all vectors $\{\boldsymbol{x}^I - \boldsymbol{x}^K | \boldsymbol{x}^I, \boldsymbol{x}^K \in \mathcal{X}_i\}$. Let $M_i \in \mathbb{R}^{N \times \binom{|\mathcal{X}_i|}{2}}$ be a matrix constructed by stacking all the $\binom{|\mathcal{X}_i|}{2}$ vectors, where $|\mathcal{X}_i|$ is the number of samples containing taxon $i$. By construction, each column of $M_i$ is the normal vector of a hyperplane orthogonal to the difference of the corresponding sample pair. (step-2): Randomly sample $(N - 1)$ hyperplanes. Choose randomly $N - 1$ columns from $M_i$. (step-3): Find the intersection of the $(N - 1)$ sampled hyperplanes to obtain an intersection line. This can be done by finding the kernel of the matrix obtained by stacking the chosen columns. Note that the randomly sampled $(N - 1)$ hyperplanes do not always intersect into a line, because some hyperplanes might be parallel with each other. However, this situation is not generic in $\mathbb{R}^N$. Thus, if the randomly sampled hyperplanes do no intersect into a line, we return to step-2 and choose a new subset of columns. (step-4): Count how many hyperplanes cross the region of the intersection line using the sign-satisfaction graph. The sign-pattern of this intersection line represent the three orthants in $\mathbb{R}^N$ crossed by all those $(N - 1)$ hyperplanes. For the remaining hyperplanes in $M_i$ (i.e., the rest of the columns in $M_i$), let $\tilde{\phi}$ be the number of those

hyperplanes that cross these three orthants. We normalize $\tilde{\phi}$ using $\phi = \tilde{\phi} / \binom{|\mathcal{X}_i|}{2}$, so that $\phi \in [0, 1]$. Notice that $\phi = 1$ means that this sign-pattern of the intersection line meets the requirements of sign-satisfaction for all the sample pairs. Therefore, the magnitude of the computed $\phi$ can be seen as the confidence of this potential solution to be the true solution of the sign-satisfaction problem. (step-5): Go back to step-2 until $\Psi \geq 1$ intersection lines have been computed.

In summary, selecting the intersection line can be seen as a 'preference' sampling in the sign-satisfaction graph, because this intersection line can be crossed by at least $(N - 1)$ hyperplanes in $M_i$. Combining the sign-satisfaction graph with the sampling procedure described above, we propose a heuristic algorithm to infer the sign-pattern of $J_i$. See Supplementary Fig. 6 for the detailed flowchart of this method.

**Limitations of the inference using relative abundance data.** High-throughput amplicon sequencing of 16S RNA has become a well-established approach for profiling microbial communities. The result of this procedure is a relative abundance profile, where the relative abundance of each taxon in the microbial community has been normalized so that their sum is one. This implies that an increase of the relative abundance of one taxon must be accompanied by a decrease of the relative abundance of other taxa.

This compositionality of relative abundance data severely limits the application of system identification methods based on temporal data, as discussed with details in refs. [15,19,59]. It also limits our method based on steady-state data. Consider, for example, that there exist three relative abundance profiles containing taxon $i$, say $\bar{\boldsymbol{x}}^I, \bar{\boldsymbol{x}}^J, \bar{\boldsymbol{x}}^K$. Since they are relative abundances, the sum of each of these samples must equal 1, implying that $\text{sum}(\bar{\boldsymbol{x}}^I - \bar{\boldsymbol{x}}^J) = \text{sum}(\bar{\boldsymbol{x}}^I - \bar{\boldsymbol{x}}^K) = \text{sum}(\bar{\boldsymbol{x}}^J - \bar{\boldsymbol{x}}^K) = 0$. Consequently, the vector $\boldsymbol{1} = (1, \ldots, 1) \in \mathbb{R}^N$ satisfies $\boldsymbol{1} \cdot (\bar{\boldsymbol{x}}^I - \bar{\boldsymbol{x}}^J) = \boldsymbol{1} \cdot (\bar{\boldsymbol{x}}^I - \bar{\boldsymbol{x}}^K) = \boldsymbol{1} \cdot (\bar{\boldsymbol{x}}^J - \bar{\boldsymbol{x}}^K) = \boldsymbol{0}$. In other words, this vector $\boldsymbol{1}$ is always orthogonal to all sample differences and the intersection line of the $(N - 1)$ hyperplanes generated by relative abundance is always $\boldsymbol{1}$. Therefore, the heuristic algorithm fails in correctly inferring the sign-pattern of Jacobian matrix using relative abundances, because it always predicts that one possible sign-pattern is $\text{sign}(\boldsymbol{s}) = (+, \ldots, +)$.

**Data availability.** All the experimental data,sets analyzed in this study are either publicly available or kindly provided by the original authors. Other data that support the findings of this study are available from the corresponding author on reasonable request.

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

## Acknowledgements

This work is supported in part by the John Templeton Foundation (Award number 51977). We thank Drs. Gabe Billings and Brigid Davis for insightful comments on the manuscript. We thank Drs. Joseph Nathaniel Paulson, Michael T. Mee, Harris H. Wang, Francesco Carrara, and Carsten F. Dormann for kindly providing their experimental data sets. We thank Dr. Liang Tian for discussions.

## Author contributions

Y.-Y.L conceived the project. Y.-Y.L and M.T.A. designed the project. Y.X. and M.T.A. did the analytical calculations. Y.X. did the numerical simulations and analyzed the empirical data. All authors analyzed the results. Y.-Y.L., Y.X., and M.T.A. wrote the manuscript. J.F, M.K.W., and S.T.W. edited the manuscript.
