## [Peer Review File · Nature Communications]

Reviewers' comments:

Reviewer #1 (Remarks to the Author):

Xiao et al describe an exciting new method for inferring interaction networks in microbial communities. There has been great excitement around using longitudinal data following a perturbation to infer interactions between microbes in complex communities such as the human gut (i.e. to parameterize Generalized Lotka-Volterra models). The current method is able to make similar inferences, but uses single time point samples from many individuals and does not require perturbation. The method uses changes of absolute taxa abundance between samples with different composition to identify the signs of interactions. If the distribution of abundances is consistent with Generalized Lotka-Volterra expectations it is further possible to predict the strength of interactions and intrinsic growth rates (but signs can be obtained even if the system deviates from GLV). The authors demonstrate that they can accurately predict interaction terms for systems simulated with a variety of underlying dynamic models. Accuracy remains above 60% even in the presence of simulated sampling bias, and the number of samples required for accurate estimates increases linearly with system complexity. Finally the authors demonstrate that inferred interactions in a system with 8 species, match the interactions determined from monoculture and pair-wise experiments in the laboratory. This inference method described will provide a useful tool for those interested in understand microbial ecology and predicting community dynamics.

One concern about the approach is that it assumes that the sign of interactions do not change between environments. This assumption has been shown not to hold in many laboratory experiments. The authors acknowledge this point, but assert that violation of the assumption requires dramatic changes in nutrient availability and therefore is less likely in systems such as the human gut. The method also assumes that systems are in approximate steady state. This assumption can be relaxed somewhat by utilizing methods that deal with noisy data, however the method would still struggle with systems that are in frequent flux. There are many cases in which both assumptions will be violated, but the authors acknowledge these constraints. Further the approach is likely to generate useful hypotheses which can subsequently be tested about the interaction networks in different communities.

An additional concern about the manuscript is its accessibility to a broad audience. While the manuscript is clearly written it appears geared toward an audience with specialty in ecological modeling and inference. That being said I think the method would be broadly employed.

Specific comments:

1) "MC" instead of "microbial community" seems like an unnecessary abbreviation that reduces clarity.

2) Further discussion of how the current method relates to other methods for inferring interactions would be useful. For example co-occurrence data has certainly previously been used to infer interactions. Additionally, it will be exciting to see how this method's inferences about interactions in the gut stack up against inferences based on longitudinal studies. I assume that manuscript is in the works however.

Page 6 - "eights orthants" should be "eight orthants".

Page 8 - "...or measure their abundance with noise" is an awkward phrase. Maybe replace with "...or there is noise in abundance measures"

Page 11 (and elsewhere) - less samples should be fewer samples

Figure 3 - It might help with clarity if words in addition to symbols could be used to label graph axes.

Reviewer #2 (Remarks to the Author):

This manuscript proposes a method to infer the interaction types between microbial species from steady state measurements of their populations. This is an important problem, and the proposed method is mathematically elegant. Unfortunately the domain of applicability/usefulness of the method is not sufficiently established. I will elaborate on specific points and suggest potential remedies.

1. The manuscript states that "the existing methods for inferring the ecological networks of microbial communities are all based on temporal abundance data". This is not quite correct. There are methods that are based on steady state measurements, such as in the article Claussen et al, PLOS Computational Biology 2017. This method also does not require a parameterized population dynamics model. Instead, it uses a Boolean (logical) model. Indeed, Boolean models have been successfully used to model ecological communities, see for example Campbell et al. PNAS 2011, and to infer microbiome networks, as in Steinway et al. 2015.

2. The adopted modeling framework, embodied in equation 1, implements the assumption that "without external perturbations those initially absent or later extinct taxa will never be present in the microbial community again as time goes by". This is only true if (i) the system is closed and (ii) the measurements are so precise that the concentration of zero can be established without a doubt. These assumptions need to be discussed and justified.

3. Application of the method involves the assumption that there is a single steady state for each species composition. This seems to be supported in the example with eight species of soil bacteria, but is it generally true? This assumption puts strong constraints on the allowed network structures (specifically, on the existence of positive cycles).

4. In figure 5b it is stated that the color depth corresponds to the relative yield values shown in (a). But there are several different kinds of exceptions. The diagonal elements the the most intense blue in b, while they are white in a, the undetermined entries have a grey, depth-free background, and the color is the opposite for the incorrect inferences. While there may be a good reason for all of these exceptions, when taken together they obstruct the point. A three color, depth-free scheme will work better.

5. The manuscript ends by saying that this method holds great promise for inferring the ecological networks of large and complex microbial communities. Some of this promise should be fulfilled in the current manuscript. The data analyzed in the Claussen et al. article (822 human microbiome compositions) would be an ideal test set of the methods proposed here, and comparison with the previously obtained results would be highly informative.

Reviewer #3 (Remarks to the Author):

The manuscript by Xiao et al. describes a procedure for inferring qualitative features of species' pairwise interactions (namely, their sign) from examples of the equilibria of the ecological network ("steady-state samples"). The work is very clearly written and organized, with excellent visuals that

clearly communicate the approach and results. The SI is written with a very impressive level of care. I particularly appreciated that the authors included a discussion of the fact that many experimental methods only measure relative abundance of species, and the difficulty this presents for the authors' method.

All in all, this is a very solid and well-written manuscript; however, as I explain below, I am not sure that Nature Communications is the most appropriate platform. I do think it would be an excellent contribution to, say, PLoS Comb Bio.

The theoretical idea presented in this work is very simple. Briefly, if I assume that the effect of species on each other is context-independent, then comparing equilibria consisting of different subsets of species should allow me to "read off" the signs of interactions. For instance, if one equilibrium differs from another only by addition of a species X, and adding X brings down the abundance of Y, then I conclude X depresses Y.

The authors' contribution is three-fold. First, they show how this very simple argument can be naturally extended to a multi-species case. Next, the paper explains how the theoretical idea can be implemented in a computationally reasonable way (including a very interesting discussion of how one might expect the necessary number of samples to scale with the problem size), and finally, the method is applied to a dataset from a synthetic community.

The explicit and unjustified assumption of context-independent interactions, however, is extremely worrying. The authors stress that their approach has the advantage of inferring interactions "without specifying any population dynamics model". But although they are indeed not relying on any single parametrized model a la generalized Lotka-Volterra, calling their approach model-independent would be a gross overstatement. It works for very narrow class of models - one so rigid that the local linearized structure around one equilibrium holds true for all states of the system. This is a very strong assumption, and undermines the authors' emphasis that "unlike the previous methods, [ours] does not require any perturbations applied to the system nor sufficiently close steady-states". This may be true, but only because they in fact assume a much stronger statement, namely that in some sense all steady states can be considered "close" (the sign structure of the Jacobian never changes, not just under a small deviation, but under a deviation of any size).

If such an assumption is adopted, it is certainly not surprising one can infer the signs of interactions. But why is this ok to assume? It is certainly NOT true that it holds for all known models - e.g. MacArthur's resource competition model is not in this class (the effect on X of adding species Y depends on the context; it might be worth stressing that this model is very much of the form (S1)). Could it be that this complication does not arise for real communities? That would definitely be interesting - and run counter to the intuition of the whole field, where most people believe that ecology is all about multistability and the absence of steady state. So showing that the authors' method often works well on real data would be a very surprising finding - but the present manuscript provides no real evidence of that. The authors apply their approach to one dataset from a recent study on a synthetic community by Friedman et al. But that work made it into Nature Eco & Evo precisely because that dataset was so surprising, and surprising in this very aspect. It presented a system where pairwise interactions were largely sufficient to predict the outcome of multi-species competition - surely we do not believe this to generally be the case? One could argue that this is precisely the one example where the authors' method can be expected to work - but until I see an application to more representative data, I remain unconvinced it could be useful in a wider practice.

Continuing to develop the method presented by the authors (by which I mean, applying it to data) could very well uncover very interesting signatures in the data. Such findings would justify the method

and make it compelling / impactful. At the moment, I think this work constitutes an invitation to follow this road (and as such, definitely publishable), but not yet a finding to report to a broad audience.

Criticizing a study for only applying a theory to one dataset rather than ten is often unfair. However, extraordinary claims require extraordinary evidence. In the Friedman et al. experimental paper the claim was: "look, assembly rules can sometimes be surprisingly simple". To justify this claim, showing one experimental example was sufficient. Here, that single example (!) is generalized to the statement that multistability is likely "really rare in real microbial communities", and cited as grounds for expecting a simplistic assumption to be widely applicable to real data.

I should stress once again that I think this is very solid and substantive manuscript, one I very much enjoyed reading. But the format/mission of Nature Communications calls for generality which in this case would be over-stated. I think it would make for a high-quality PLoS Comp Bio paper, where it could be basically published "as is", after toning down the statements of model-independence. The fact that a significant portion of the main text is dedicated to a discussion of algorithm performance also suggests that a comp-bio journal might be more appropriate (brute force vs heuristic; why it is necessary and how well it performs; minimal required sample size; benchmarking on simulated data; the detailed Fig. 3).

A few minor comments:

1) "which is undirected and do not encode"  "and DOES not encode"

2) "such MCs often display highly intrinsic stability and resilience" -- I don't understand the phrase "highly intrinsic"

3) and passes the origin. -> and passes through the origin?

4) "The regions (including the origin and two quadrants) crossed by this green line provide the minimum set of possible sign-patterns" -- I do not understand in what sense this is a "minimum set". The authors probably mean "minimal", but even so, the meaning is unclear. The phrase "maximal set" seems equally appropriate, perhaps even more so?

5) "There are three possible steady-state samples" (Fig 1b); "seven possible steady-state samples" (Fig. 1d) -- I think the phrasing here needs to be more careful. These statements refer to a classification of states based on presence-absence of species. One could e.g. say there are three/seven possible TYPES of equilibria. It is certainly not true that all three/seven are "possible" in the sense that they need not exist for a given system (e.g. a mutually repressive network cannot support coexistence). Further, it is not clear to me that there can be at most 1 steady state in each class. Thus the statement that "there are 7 possible steady-state SAMPLES" is incorrect and needs to be phrased more carefully.

RESPONSE to Reviewer #1

Xiao et al describe an exciting new method for inferring interaction networks in microbial communities. There has been great excitement around using longitudinal data following a perturbation to infer interactions between microbes in complex communities such as the human gut (i.e. to parameterize Generalized Lotka-Volterra models). The current method is able to make similar inferences, but uses single time point samples from many individuals and does not require perturbation. The method uses changes of absolute taxa abundance between samples with different composition to identify the signs of interactions. If the distribution of abundances is consistent with Generalized Lotka-Volterra expectations it is further possible to predict the strength of interactions and intrinsic growth rates (but signs can be obtained even if the system deviates from GLV). The authors demonstrate that they can accurately predict interaction terms for systems simulated with a variety of underlying dynamic models. Accuracy remains above 60% even in the presence of simulated sampling bias, and the number of samples required for accurate estimates increases linearly with system complexity. Finally, the authors demonstrate that inferred interactions in a system with 8 species, match the interactions determined from monoculture and pair-wise experiments in the laboratory. This inference method described will provide a useful tool for those interested in understand microbial ecology and predicting community dynamics.

We thank Reviewer #1 for his/her very effective summary of our paper, and his/her very positive assessment of the novelty, general interest and potential applications of our method.

One concern about the approach is that it assumes that the sign of interactions does not change between environments. This assumption has been shown not to hold in many laboratory experiments. The authors acknowledge this point, but assert that violation of the assumption requires dramatic changes in nutrient availability and therefore is less likely in systems such as the human gut.

We thank Reviewer #1 for this very legitimate concern. Indeed, if the interaction types (signs) change between environments, then the inference of interaction types becomes an ill-defined problem, simply because we will have a “moving target” and different subsets of steady-state samples will offer different interaction types. Since other reviewers raised similar concern, we address it all together in the end this rebuttal letter (see Pages 11-13, as well as Figs. R1-R2).

The method also assumes that systems are in approximate steady state. This assumption can be relaxed somewhat by utilizing methods that deal with noisy data, however the method would still struggle with systems that are in frequent flux. There are many cases in which both assumptions will be violated, but the authors acknowledge these constraints.

We thank Reviewer #1 for this very insightful comment. As summarized nicely in the first paragraph of Reviewer #1’s report, our method is based on comparing the absolute abundances of species in different steady-state samples. For systems that are in frequent flux, where steady-state samples are hard to collect, our method is not applicable. In the revised manuscript, we explicitly mentioned this point (see the 3rd paragraph of the Discussion section).

Further the approach is likely to generate useful hypotheses which can subsequently be tested about the interaction networks in different communities.

We fully agree with Reviewer #1. In the revised manuscript, we analyzed a synthetic community of maize roots with 7 bacterial species [1]. The dataset consists of 7 sextets (i.e., 7 steady-state samples involving 6 of the 7 species) and 1 septet (i.e., a steady-state sample involving all the 7 species). We inferred the interaction types (i.e., the sign-pattern of the Jacobian matrix) using only the 7 sextets, finding that the inferred interaction types can predict the species abundance changes before and after respectively adding one species into each of the 7 sextets with an accuracy ~ 71.43%. Please see Page 14 of the rebuttal letter and Fig. R4 for details.

An additional concern about the manuscript is its accessibility to a broad audience. While the manuscript is clearly written, it appears geared toward an audience with specialty in ecological modeling and inference. That being said I think the method would be broadly employed.

We thank Reviewer #1 for this very constructive comment. In the revised manuscript, we have paid special attention to the wording of the main text to improve its accessibility to broader audience. In particular, in the 3rd paragraph of the Introduction section of the revised main text, we offer a simple explanation of the basic idea of our method, which should be accessible to a broad audience.

Specific comments:

1) "MC" instead of "microbial community" seems like an unnecessary abbreviation that reduces clarity.

We have replaced "MC" by "microbial community" throughout the revised manuscript.

2) Further discussion of how the current method relates to other methods for inferring interactions would be useful. For example, co-occurrence data has certainly previously been used to infer interactions. Additionally, it will be exciting to see how this method's inferences about interactions in the gut stack up against inferences based on longitudinal studies. I assume that manuscript is in the works however.

We fully agree with Reviewer #1. In the revised manuscript, we offered detailed discussion on the difference between our method and existing ones. Regarding the comparison of our inference results of human gut microbiome with existing results based on temporal data, this is still an ongoing study and may take a while to get enough samples with absolute abundances. We feel this is beyond the scope of the current project, and will leave it as a future work.

3) Page 6 - "eights orthants" should be "eight orthants".

Typo fixed.

4) Page 8 - "...or measure their abundance with noise" is an awkward phrase. Maybe replace with "...or there is noise in abundance measures".

We have revised that sentence accordingly.

5) Page 11 (and elsewhere) - less samples should be fewer samples.

Fixed.

6) Figure 3 - It might help with clarity if words in addition to symbols could be used to label graph axes.

We have improved the figure presentation accordingly.

Finally, we thank Reviewer #1 again for his/her very insightful and constructive comments. We hope our responses above have addressed those very legitimate issues/concerns in a satisfactory manner.

Response to Reviewer #2

This manuscript proposes a method to infer the interaction types between microbial species from steady state measurements of their populations. This is an important problem, and the proposed method is mathematically elegant. Unfortunately, the domain of applicability/usefulness of the method is not sufficiently established. I will elaborate on specific points and suggest potential remedies.

We thank Reviewer #2 for his/her very positive assessment of the importance of the problem and the mathematical elegance of our method. We next address each of the reviewer's concerns in order.

1. The manuscript states that "the existing methods for inferring the ecological networks of microbial communities are all based on temporal abundance data". This is not quite correct. There are methods that are based on steady state measurements, such as in the article Claussen et al, PLOS Computational Biology 2017. This method also does not require a parameterized population dynamics model. Instead, it uses a Boolean (logical) model. Indeed, Boolean models have been successfully used to model ecological communities, see for example Campbell et al. PNAS 2011, and to infer microbiome networks, as in Steinway et al. 2015.

We thank Reviewer #2 for pointing out those relevant references, which have been briefly discussed and cited in the revised manuscript.

We emphasize that in the statement "*the existing methods for inferring the ecological networks of microbial communities are all based on temporal abundance data*", by "*ecological network*" we mean a **directed** graph that depicts the direct ecological interactions between species, rather than an **undirected** graph representing indirect (or effective) ecological interactions inferred from co-occurrence data. We apologize for not making the point clear enough in the previous version of the manuscript. In the revised version, we explicitly mention this point in the 1st paragraph of the Introduction section.

In Claussen et al, PLOS Computational Biology 2017, the authors presented an interesting method based on Boolean operations applied to microbial co-occurrence data. The resulting species interaction network is **undirected**, which is fundamentally different from the **directed** ecological network inferred by our method using steady-state data or other methods using Boolean model and time-series data (such as Campbell et al, PNAS 2011; Steinway et al, PLOS Computational Biology 2015).

2. The adopted modeling framework, embodied in equation 1, implements the assumption that "without external perturbations those initially absent or later extinct taxa will never be present in the microbial community again as time goes by". This is only true if (i) the system is closed and (ii) the measurements are so precise that the concentration of zero can be established without a doubt. These assumptions need to be discussed and justified.

We thank Reviewer #2 for this very insightful comment. Indeed, the microbial community can be invaded by other species that are currently absent. If this migration occurs relatively fast, then our modeling framework will not work. In the revised manuscript, we made this point clearly (see main text Page 5, first paragraph).

Regarding the precise measurement of zero concentration, in the previous version of our manuscript, we have demonstrated the robustness of our method in the presence of measurement noise (see main text Fig. 3b, Fig. 4c-f). We found that with a reasonable level of measurement noise, our inference method still works very well.

3. Application of the method involves the assumption that there is a single steady state for each species composition. This seems to be supported in the example with eight species of soil bacteria, but is it generally true? This assumption puts strong constraints on the allowed network structures (specifically, on the existence of positive cycles).

We thank Reviewer #2 for this very insightful comment. Indeed, true multi-stability may exist for certain microbial communities, which will cause trouble in inferring the interaction types (signs). In the revised manuscript, we rigorously proved that if the sign-pattern of the Jacobian matrix is constant, then there is no true multi-stability. Equivalently, if a community displays true multi-stability, then the sign-pattern of the Jacobian matrix is not constant (see Proposition 1 in Supplementary Note 1.4 for the proof).

In practice, we can detect the presence of true multi-stability for a collection of steady-state samples, hence falsifying the assumption of constant interaction types. We emphasize that by introducing a criterion to falsify our assumption, we actually enhance the applicability of our method.

4. In figure 5b it is stated that the color depth corresponds to the relative yield values shown in (a). But there are several different kinds of exceptions. The diagonal elements the the most intense blue in b, while they are white in a, the undetermined entries have a grey, depth-free background, and the color is the opposite for the incorrect inferences. While there may be a good reason for all of these exceptions, when taken together they obstruct the point. A three color, depth-free scheme will work better.

We thank Reviewer #2 for this very constructive comment. We have revised the color scheme of Figure 5b accordingly.

5. The manuscript ends by saying that this method holds great promise for inferring the ecological networks of large and complex microbial communities. Some of this promise should be fulfilled in the current manuscript. The data analyzed in the Claussen et al. article (822 human microbiome compositions) would be an ideal test set of the methods proposed here, and comparison with the previously obtained results would be highly informative.

We thank Reviewer #2 for this suggestion. Unfortunately, we cannot apply our method to the data analyzed in the Claussen et al. article. The data analyzed in that paper are relative abundances,

while our method requires absolute abundances (see Supplementary Note 2.4 for the detailed explanation). Note that the compositionality of relative abundance data also represents a major challenge for other methods in inferring directed ecological networks based on time-series data [2,3].

We also emphasize that, besides the existing dataset [4], in the revised manuscript we analyzed three additional datasets: (i) a synthetic microbial community of two cross-feeding partners with different amount of resource availability [5]; (ii) a synthetic community of maize roots with 7 bacterial species [1]; and (iii) synthetic communities of *Escherichia coli* bacteria of increasing complexity to measure general properties enabling metabolic exchange of amino acids [6]. Since other reviewers also raise similar concerns about the application of our method to other datasets, we address these all together in the end of this rebuttal letter (see Page 14-15, and Figs. R3-R5).

Finally, we thank Reviewer #2 again for his/her very insightful and constructive comments. We hope our responses above have addressed those very legitimate issues/concerns in a satisfactory manner.

Response to Reviewer #3

The manuscript by Xiao et al. describes a procedure for inferring qualitative features of species' pairwise interactions (namely, their sign) from examples of the equilibria of the ecological network ("steady-state samples"). The work is very clearly written and organized, with excellent visuals that clearly communicate the approach and results. The SI is written with a very impressive level of care. I particularly appreciated that the authors included a discussion of the fact that many experimental methods only measure relative abundance of species, and the difficulty this presents for the authors' method.

All in all, this is a very solid and well-written manuscript; however, as I explain below, I am not sure that Nature Communications is the most appropriate platform. I do think it would be an excellent contribution to, say, PLoS Comb Bio.

We thank Reviewer #3 for his/her very positive assessment of the presentation of our work. Next, we address each point raised by the Reviewer in order.

The theoretical idea presented in this work is very simple. Briefly, if I assume that the effect of species on each other is context-independent, then comparing equilibria consisting of different subsets of species should allow me to "read off" the signs of interactions. For instance, if one equilibrium differs from another only by addition of a species X, and adding X brings down the abundance of Y, then I conclude X depresses Y.

The authors' contribution is three-fold. First, they show how this very simple argument can be naturally extended to a multi-species case. Next, the paper explains how the theoretical idea can be implemented in a computationally reasonable way (including a very interesting discussion of how one might expect the necessary number of samples to scales with the problem size), and finally, the method is applied to a dataset from a synthetic soli community.

Indeed, the basic idea of our method is very simple. We thank Reviewer #3 for his/her very precise description of the essence of our method, which helps us a lot improve the accessibility of our paper for a broader audience (as suggested by Reviewer #1). We also highly appreciate Reviewer #3's very clear summary of our three-fold contributions.

The explicit and unjustified assumption of context-independent interactions, however, is extremely worrying. The authors stress that their approach has the advantage of inferring interactions "without specifying any population dynamics model". But although they are indeed not relying on any single parametrized model like generalized Lotka-Volterra, calling their approach model-independent would be a gross overstatement. It works for very narrow class of models - one so rigid that the local linearized structure around one equilibrium holds true for all states of the system. This is a very strong assumption, and undermines the authors' emphasis that "unlike the previous methods, [ours] does not require any perturbations applied to the system nor sufficiently close steady-states". This may be true, but only because they in fact assume a much stronger statement, namely that in some sense all steady states can be considered "close" (the sign structure of the Jacobian never changes, not just under a small deviation, but under a deviation of any size). If such an assumption is adopted, it is certainly not surprising one can infer the signs of interactions. But why is this ok to assume? It is certainly NOT true that it holds for all known models - e.g. MacArthur's resource competition model is not

in this class (the effect on X of adding species Y depends on the context; it might be worth stressing that this model is very much of the form (S1)).

We thank Reviewer #3 for this very legitimate concern. We fully agree with the reviewer that the context-independent interactions (or the constant sign-pattern of the Jacobian matrix: $\text{sign}(J) = \text{const}$) is a strong assumption and is not true for all known models. Since other reviewers also raise similar concern regarding the context-independent interactions, we address it all together in the end of this rebuttal letter (see Page 11-13, as well as Figs. R1-R2). Basically, in the revised manuscript, we introduced a criterion to falsify the assumption that $\text{sign}(J) = \text{const}$. In case $\text{sign}(J) \neq \text{const}$, our method can still infer the zero of J , representing the structure of the ecological network. Moreover, in case $\text{sign}(J) \neq \text{const}$ but the steady-state samples were still collected from the microbial community under the same or very similar environment, we show that we can interpret our inferred $\text{sign}(J)$ as the overall or effective impact between different species during the transitions between steady states.

Could it be that this complication does not arise for real communities? That would definitely be interesting - and run counter to the intuition of the whole field, where most people believe that ecology is all about multistability and the absence of steady state. So showing that the authors' method often works well on real data would be a very surprising finding - but the present manuscript provides no real evidence of that. The authors apply their approach to one dataset from a recent study on a synthetic community by Friedman et al. But that work made it into Nature Eco & Evo precisely because that dataset was so surprising, and surprising in this very aspect. It presented a system where pairwise interactions were largely sufficient to predict the outcome of multi-species competition - surely we do not believe this to generally be the case? One could argue that this is precisely the one example where the authors' method can be expected to work - but until I see an application to more representative data, I remain unconvinced it could be useful in a wider practice.

We thank Reviewer #3 for this comment. Since other reviewers also raise similar concerns about the applicability of our method to other datasets, we address it all together in the end of this rebuttal letter (see Page 14-15, as well as Figs. R3-R5). We emphasize that, besides the existing dataset of Friedman et al. [4], in the revised manuscript we also analyzed three additional datasets: (i) a synthetic microbial community of two cross-feeding partners with different amount of resource availability [5]; (ii) a synthetic community of maize roots with 7 bacterial species [1]; and (iii) synthetic communities of *Escherichia coli* bacteria of increasing complexity to measure general properties enabling metabolic exchange of amino acids [6].

Continuing to develop the method presented by the authors (by which I mean, applying it to data) could very well uncover very interesting signatures in the data. Such findings would justify the method and make it compelling/impactful. At the moment, I think this work constitutes an invitation to follow this road (and as such, definitely publishable), but not yet a finding to report to a broad audience.

Criticizing a study for only applying a theory to one dataset rather than ten is often unfair. However, extraordinary claims require extraordinary evidence. In the Friedman et al. experimental paper the claim was: "look, assembly rules can sometimes be surprisingly simple".

To justify this claim, showing one experimental example was sufficient. Here, that single example (!) is generalized to the statement that multistability is likely "really rare in real microbial communities", and cited as grounds for expecting a simplistic assumption to be widely applicable to real data.

We thank Reviewer #3 for this critical comment. We agree with Reviewer #3 that the sentence "Note that true multi-stability was observed in only one of the 101 species combinations, suggesting that it is really rare in real microbial communities." is quite misleading. In the revised manuscript, it has been removed.

Note that our method actually heavily leverages the fact that a microbial community of N taxa could have many different steady-state samples associated with different species collections. Those steady-state samples collected in experiments can be considered as *alternative stable states* of the system. Any two of them that share a common species i can be used to infer the sign-pattern of J_i , i.e., the i -th row of the Jacobian matrix.

As mentioned above, in the revised manuscript we have carefully justified our assumption of context-independent interactions by introducing a simple criterion to falsify it using steady-state samples (based on the detection of *true multi-stability*). Also, we have demonstrated that even if interactions are context-dependent, i.e., $\text{sign}(J) \neq \text{const}$, we can still infer the zero of J , which represents the structure of the ecological network and is interesting by itself. We also demonstrated the application of our method in three additional real datasets. We hope Reviewer #3 now find that our method is compelling enough to report to a broad audience.

I should stress once again that I think this is very solid and substantive manuscript, one I very much enjoyed reading. But the format/mission of Nature Communications calls for generality which in this case would be over-stated. I think it would make for a high-quality PLoS Comp Bio paper, where it could be basically published "as is", after toning down the statements of model-independence. The fact that a significant portion of the main text is dedicated to a discussion of algorithm performance also suggests that a comp-bio journal might be more appropriate (brute force vs heuristic; why it is necessary and how well it performs; minimal required sample size; benchmarking on simulated data; the detailed Fig. 3).

We thank Reviewer #3 again for his/her overall positive assessment of our work. Regarding *Nature Communications* vs. *PLoS Computational Biology*, we think that the former is a much better venue to publish our work based on the following considerations:

- We believe our method offers a novel framework to infer microbial interactions and reconstruct ecological networks, and hence represents a key step towards reliable modeling of complex, real-world microbial communities.
- We share the same opinion as Reviewer #1 that our method would be broadly employed. In fact, we anticipate that our work will be of immediate relevance and impact for a rather broad readership beyond computational biology. *First*, for **experimental microbiologists** studying the host-associated or environmental microbial communities, our method will prompt them to rethink the inference of ecological networks using steady-state data, rather

than designing perturbations to obtain more informative time series data. *Second*, our results will help **theoretical ecologists** with dynamics systems or system identification backgrounds better understand the challenges of inferring population dynamics and ecological networks, and may inspire them to develop new methods to better leverage existing steady-state data in other ecological systems. *Third*, our steady-state based network inference will inspire **control theorists** and **network scientists** to develop similar methods in mapping the underlying network of general complex dynamical systems.

A few minor comments:

1) "which is undirected and do not encode"  "and DOES not encode"

Typo fixed.

2) "such MCs often display highly intrinsic stability and resilience" -- I don't understand the phrase "highly intrinsic"

We have deleted the phrase "highly intrinsic" in that sentence.

3) and passes the origin. -> and passes through the origin?

We have revised that sentence accordingly.

4) "The regions (including the origin and two quadrants) crossed by this green line provide the minimum set of possible sign-patterns" -- I do not understand in what sense this is a "minimum set". The authors probably mean "minimal", but even so, the meaning is unclear. The phrase "maximal set" seems equally appropriate, perhaps even more so?

Sorry for not making this statement clear enough in the previous version. In the revised manuscript, we have removed the word "minimum".

5) "There are three possible steady-state samples" (Fig 1b); "seven possible steady-state samples" (Fig. 1d) -- I think the phrasing here needs to be more careful. These statements refer to a classification of states based on presence-absence of species. One could e.g. say there are three/seven possible TYPES of equilibria. It is certainly not true that all three/seven are "possible" in the sense that they need not exist for a given system (e.g. a mutually repressive network cannot support coexistence). Further, it is not clear to me that there can be at most 1 steady state in each class. Thus the statement that "there are 7 possible steady-state SAMPLES" is incorrect and needs to be phrased more carefully.

We thank Reviewer #3 for this very insightful comment. In the revised manuscript, we have rephrased those sentences carefully.

Finally, we thank Reviewer #3 again for his/her very insightful and constructive comments. We hope our responses above have addressed those very legitimate issues/concerns in a satisfactory manner.

Responses to common concerns from reviewers

1. Regarding our assumption that the ecological interaction types do not vary across the observed steady-state samples.

In the previous version of our manuscript, we made an explicit assumption that the nature of the ecological interactions (i.e., promotion, inhibition, or neutral) between any two taxa does not vary across the observed steady-state samples, though their interaction strengths might change. Mathematically, this is equivalent to assume that the sign-pattern of the Jacobian matrix is constant over those observed samples. Note that the magnitude of the Jacobian matrix may still vary over samples, and we only assume that its sign-pattern is constant. Otherwise, if the sign-pattern of the Jacobian matrix is not constant, its inference will be an ill-defined problem because we have a “moving target” and different subsets of samples will provide different answers.

Under the same culture conditions such as similar nutrient availability and uniform spatial structures, we think this assumption will very likely hold [5,7]. We admit that in general the nature of inter-taxa interactions may change, for example, depending on the nutrient availability. To extend our method, in the revised manuscript we made the following progresses:

(1) In case the sign-pattern of the Jacobian matrix is not constant, but the steady-state samples were still collected from the microbial community under the same or similar environmental conditions (e.g., nutrient availability), we can infer the zero of the Jacobian matrix (see the revised manuscript Supplementary Note 1.3 for the proof, and Supplementary Note 3 for simulation results). Note that the zero of the Jacobian matrix (i.e., $J_{ij} = 0$ or $J_{ij} \neq 0$) determines the topology of the ecological network of the microbial community, which is interesting by itself and can be very useful in control theoretical analysis [8].

To demonstrate this point clearly, let’s consider a toy model with two taxa:

$$\begin{cases} \dot{x}_1 = x_1(0.5 - x_1 + 0.1x_2), \\ \dot{x}_2 = x_2(x_2 - 0.6)(0.2 - x_2), \end{cases} \quad (\text{R1})$$

where the Jacobian matrix is

$$J = \begin{bmatrix} -1 & 0.1 \\ 0 & 0.8 - 2x_2 \end{bmatrix}.$$

Note that the sign of J_{22} depends on the value of x_2 . Fig. R1b shows that J_{22} indeed changes its sign from positive to negative during the growth process (Fig. R1a).

In the absence of measurement noise (Fig. R1c,d), we can successfully infer the zero of J . Fig. R1c shows that according to the position of $\mathbf{x}^{\{1,2\}} - \mathbf{x}^{\{1\}}$, the green line that is orthogonal to the red line cannot yield a zero entry for \mathbf{J}_1 , implying that $J_{11} \neq 0$ and $J_{12} \neq 0$. This is consistent with the ground truth. Fig. R1d shows that for \mathbf{J}_2 , the green line that is orthogonal to the blue line locates on the axis of x_2 , implying that $J_{21} = 0$ and $J_{22} \neq 0$, which is also consistent with the ground truth.

In the presence of measurement noise (Fig. R1e,f), the angle between the x_1 -axis (or the x_2 -axis) and the green line can be used to determine if $J_{ij} = 0$ or not. Similarly, as we did in the main text, here we add noise to each non-zero entry x_i^l of a steady-state sample \mathbf{x}^l by replacing x_i^l with $x_i^l + \eta u$, where $u \sim U[-x_i^l, x_i^l]$ is a random number uniformly distributed in the interval $[-x_i^l, x_i^l]$ and

$\eta \geq 0$ quantifies the noise level. When the noise level $\eta = 0.1$, in Fig. R1e the angle between the x_1 -axis and green line is large enough and we can safely conclude that $J_{12} \neq 0$. However, in Fig. R1f the green line deviates only slightly from the x_2 -axis and the deviation (to the left or right of the x_2 -axis) displays stochastic nature over different measurements. If we choose a threshold value θ , and if the absolute value of the average deviation angle over different measurements is smaller than θ , we can conclude that $J_{21} = 0$. We acknowledge that this will sometimes consider very weak interactions as zero in the inferred topology, but at least it offers a pragmatic approach to infer strong interactions. See Supplementary Note 3 for more results.

(2) In case the sign-pattern of the Jacobian matrix is not constant, but the steady-state samples were still collected from the microbial community under the same or similar environmental conditions (e.g., nutrient availability), we can interpret our inferred sign-pattern as the overall impact (positive, negative, or neutral) between different taxa during the transitions between steady states. Note that regardless of the sign-pattern of the Jacobian matrix is constant or not, our method can correctly infer the sign of $\int_0^1 J_{ij}(\mathbf{x}^l + \sigma(\mathbf{x}^k - \mathbf{x}^l)) d\sigma$, which reflects an overall impact (positive, negative or neutral) of taxon j on taxon i during the transition from the steady state \mathbf{x}^l ($\sigma = 0$) to \mathbf{x}^k ($\sigma = 1$).

To demonstrate this point more clearly, let's consider a toy model of two species X and Y [5]. Each species has a per capita growth rate that is modulated by its mutualistic partner as well as the resource. The population dynamics of this toy model is given by

$$\begin{cases} \dot{X} = r_x X \left(\frac{Y+a}{Y+a+\kappa} \right) (1 - X - Y) - \delta X, \\ \dot{Y} = r_y Y \left(\frac{\beta X+a}{\beta X+a+\kappa} \right) (1 - X - Y) - \delta Y. \end{cases} \quad (\text{R2})$$

Here r_x and r_y are the growth rates, a is the amount of resource, δ is the death rate, κ is an effective Monod constant, and β (> 0) quantifies the asymmetry of benefit that each species receives from its partner. The elements of the Jacobian matrix of this community are given by

$$\begin{cases} J_{11} = -r_x \frac{Y+a}{Y+a+\kappa}, \\ J_{12} = r_x (1 - X - Y) \frac{\kappa}{(Y+a+\kappa)^2} - r_x \frac{Y+a}{Y+a+\kappa}, \\ J_{21} = r_y (1 - X - Y) \frac{\beta\kappa}{(\beta X+a+\kappa)^2} - r_y \frac{\beta X+a}{\beta X+a+\kappa}, \\ J_{22} = -r_y \frac{\beta X+a}{\beta X+a+\kappa}. \end{cases}$$

Note that J_{11} and J_{22} are always negative, while J_{12} and J_{21} may change their signs depending on the particular abundances of X and Y , as well as the model parameters. Indeed, this model can capture the transition between the different regimes of ecological interaction depending upon the amount of resource (a). Fig. R2a shows that there are three regimes with different overall inter-species interactions, from mutualism to parasitism and competition. Here, the overall interaction types are determined by comparing the difference of steady states between monocultures (dashed lines) and co-cultures (solid lines) to calculate the relative yield, which indicates the promotion or inhibition impact between two species. Since the Jacobian matrix of this model may change its

sign-pattern over time, the sign of relative yields can be interpreted as the overall (or effective) impact between two taxa, denoted as J_{eff} (Fig. R2a).

Now we apply our method to the steady states of this model. Fig. R2b-d show the diagrams of our inference method for each regime, and the inferred overall inter-species interaction types, denoted as \tilde{J}_{eff} , are consistent with the ground truth (shown in Fig. R2a). Fig. R2e-g show the value of $J_{12}(\mathbf{x}^{\{1,2\}} + \sigma(\mathbf{x}^{\{1\}} - \mathbf{x}^{\{1,2\}}))$ and $J_{21}(\mathbf{x}^{\{1,2\}} + \sigma(\mathbf{x}^{\{2\}} - \mathbf{x}^{\{1,2\}}))$ as a function of σ corresponding to the transition between \mathbf{x}^I ($\sigma = 0$) and \mathbf{x}^K ($\sigma = 1$), and the shade areas are the value of $\int_0^1 J_{12}(\mathbf{x}^{\{1,2\}} + \sigma(\mathbf{x}^{\{1\}} - \mathbf{x}^{\{1,2\}}))d\sigma$ and $\int_0^1 J_{21}(\mathbf{x}^{\{1,2\}} + \sigma(\mathbf{x}^{\{2\}} - \mathbf{x}^{\{1,2\}}))d\sigma$. For example, when $a = 0.15$, both $\int_0^1 J_{12}(\mathbf{x}^{\{1,2\}} + \sigma(\mathbf{x}^{\{1\}} - \mathbf{x}^{\{1,2\}}))d\sigma$ and $\int_0^1 J_{21}(\mathbf{x}^{\{1,2\}} + \sigma(\mathbf{x}^{\{2\}} - \mathbf{x}^{\{1,2\}}))d\sigma$, corresponding to the shade areas in Fig. R2e, are positive. Although J_{12} and J_{21} display both negative and positive value in the whole range of σ , the positive J_{12} and J_{21} dominate in the transition between two steady states. Hence, overall species X and species Y are mutualistic to each other. Fig. R2f and Fig. R2g show the similar analysis of J_{12} and J_{21} for $a = 0.2$ and $a = 0.5$, respectively.

This example demonstrates that even if the sign-pattern of the Jacobian matrix is not constant, we can interpret our inferred sign-pattern as the overall impact (positive, negative, or neutral) between different species during the transitions between steady states, as long as those steady-state samples were still collected from the microbial community under the same environment. Of course, if the samples were collected from totally different environment (e.g., the three different regimes in Fig. R2a), then our method will fail. In fact, the inference of sign-pattern becomes ill-defined in this case. We have to focus on the inference of the zero, i.e., the ecological network structure.

(3) We can introduce a criterion to check if the sign-pattern of the Jacobian matrix is constant. In Proposition 1 of the revised Supplementary Note 1.4, we rigorously proved that if $\text{sign}(J) = \text{const}$, then there cannot be *true multi-stability*. Equivalently, if a microbial community displays *true multi-stability*, then $\text{sign}(J) \neq \text{const}$. Here, true multi-stability means that from a community of N taxa, there exists a subset of M ($\leq N$) taxa that displays multiple different steady-state abundance profiles, where all the M taxa have positive abundances and the other ($N - M$) taxa are absent. In practice, we can detect the presence of true multi-stability by examining the collected steady-state samples. If true multi-stability is detected, then we know immediately that our assumption that $\text{sign}(J) = \text{const}$ is invalid, and we should only infer the zero of J , i.e., the structure of the ecological network. If no, then at least our assumption $\text{sign}(J) = \text{const}$ is consistent with the collected steady-state samples, and we can use our method to infer $\text{sign}(J)$, i.e., the ecological interaction types. In short, by introducing a criterion to falsify our assumption, we significantly enhance the applicability of our method.

Note that to infer $\text{sign}(J)$, our method just assumes that $\text{sign}(J) = \text{const}$ (which implies that there cannot be true multi-stability), but it doesn't require the system to be globally stable (i.e., starting from different initial conditions, it always reaches the same final stable steady-state).

2. Regarding the application of our method to different datasets

Besides the dataset [4] analyzed in the previous version of our manuscript, in the revised manuscript we also analyzed three additional datasets [1,5,6]:

(1) A synthetic microbial community of two cross-feeding partners with different amount of resource availability [5]. In this community, two non-mating strains of the budding yeast, *Saccharomyces cerevisiae*, were engineered to be deficient in the biosynthesis of one of two essential amino acid tryptophan (Trp) or leucine (Leu), and to overproduce the amino acid required by their partner. It has been demonstrated that these two strains form a community with cross-feeding mutualism, where each strain provides the amino acid needed by its partner. In [5], the authors inoculated monocultures and co-cultures at a range of concentrations of supplemented amino acids in a well-mixed liquid batch. Fig. R3a-c show the abundance of the co-cultures and monocultures for the Trp and Leu strains at low, medium and high levels of supplemented amino acids. After cultivation for seven days, both strains reach steady states. Note that for each scenario a constant amount of resource was inoculated at the beginning. Here the interaction types are defined by comparing the abundance of co-cultures with monocultures at the end of cultivation. As the supply of amino acids increases from low, to medium to high concentrations, the interaction between this pair of strains shifts from obligatory mutualism (Fig. R3a) to facultative mutualism (Fig. R3b), and parasitism (Fig. R3c), respectively.

We applied our inference method to each scenario. Fig. R3d-f show the diagrams of our inference results, which are consistent with the empirical observations. For example, in Fig. R3e,f, the cyan line orthogonal to the red line is very close to the Leu axis, implying the impact of Trp on Leu is very weak. Especially in Fig. R3f, this promotion effect can be ignored.

(2) A synthetic bacterial community of maize roots [1]. There are in total 7 bacterial species (Ecl, Sma, Cpu, Opi, Ppu, Hfr and Cin) in this community. The available steady-state data consists of 7 sextets (i.e., data from seven experiments in which six different species grow together) and 1 septet (i.e., data from one experiment in which the seven species grow together). This leads to a total of 8 steady-state samples, see Fig. R4a.

First, based on our theoretical result showing that in the generalized Lotka-Volterra (GLV) model the steady states that share common species will align into a hyperplane, we concluded that this bacterial community does not follow the GLV dynamics (see Fig. R4b). Thus, we have to focus on inferring the interaction types, rather than interaction strengths.

Second, only using the 7 sextets we inferred the sign-pattern of the Jacobian matrix (Fig. R4c). Based on the inferred sign of J_{ij} , we can predict how the abundance of species- i will change, when we add species- j to the community. For example, if we add Ecl to a community consisting of the other 6 species (i.e., Sma, Cpu, Opi, Ppu, Hfr and Cin), we predict that the abundance of Sma, Opi, Ppu, Hfr and Cin will increase, while the abundance of Cpu will decrease. Note that our prediction only considers the direct ecological interactions between species and ignores the indirect impact among species. Indeed, Ecl promotes Opi, but Ecl also promotes Hfr that inhibits Opi. Hence the net effect of Ecl on Opi is hard to tell without knowing the interaction strengths. Nevertheless, we found that our prediction is consistent with experimental observation (Fig. R4d, first column).

Finally, we systematically compared our predictions of species abundance changes with experimental observations. There are in total 7 sextets (i.e., 7 steady-state samples that consists of 6 species), corresponding to the 7 columns in Fig. R4d. We add the corresponding missing species back to the community, and check the abundance changes of the existing 6 species. There are in total $6 \times 7 = 42$ abundance changes. We found that our inferred sign-pattern of the Jacobian matrix (Fig. R4c) can correctly predict 30 of the 42 abundance changes (accuracy $\sim 71.43\%$). Moreover, for those false predictions, the detailed values of the abundance changes are actually relatively small (comparing to those of correct predictions). Note that we only used 7 steady-samples to infer the interaction types. If more steady-state samples were available, we assume the prediction accuracy of our method can would be improved.

(3) Synthetic communities of *Escherichia coli* bacteria of increasing complexity to measure general properties enabling metabolic exchange of amino acids [6]. Starting from a prototrophic *E. coli* derivative MG1655, the authors of [6] generated 14 strains, each containing a gene knockout that lead to an auxotrophic phenotype of 1 of 14 essential amino acids. By convention, the authors labeled each auxotrophic strain by the amino acid it lacks. For example, the methionine auxotroph $\Delta metA$ is strain *M*. It was confirmed that the 14 auxotrophs (*C, F, G, H, I, K, L, M, P, R, S, T, W, and Y*) show no growth in M9-glucose minimal media after 4 days. Indeed, they grow only when supplemented with the essential amino acid they were not able to produce. This dataset consists of co-cultures of all 91 possible strain pairs from the 14 characterized auxotrophic strains. For each pairwise coculture, we are able to calculate the total fold growth, i.e., the yield of the community calculated by (total final cell density)/(total initial cell density), as well as the fold growth of each strain. Since these auxotrophic strains cannot grow by themselves, if strain *i* is able to grow as a co-culture when paired with strain *j*, and strain *i*'s fold growth $F_{ij} > 1$, this implies that strain *j* promotes the growth of strain *i*, i.e., $J_{ij} > 0$. By contrast, if $F_{ij} < 1$, we cannot conclusively say that $J_{ij} < 0$ because we lack the experimental monoculture data. Therefore, the fold-growth metric can only be used to detect a promotion effect between two strains.

First, we find that R^2 of all fitted hyperplanes are smaller than 0.9, implying that the population dynamics of this microbial community cannot be properly described by the GLV model (Fig. R5a). Second, we use our heuristic algorithm to infer the interaction types (Fig. R5b). Note that the complexity of the inference approaches $3^{14} \sim 4 \times 10^6$ if we use the brute-force algorithm. We find that the types of 14 pairwise interactions cannot be determined by the given dataset (marked in gray in Fig. R5b). Third, we show the fold growth matrix $F = (F_{ij})$ calculated from real data (Fig. R5c), with F_{ij} the fold growth of strain *i* (row) in the co-culture with strain *j* (column). Here we set $F_{ij} \geq 20$ as an indication of promotion effect of strain *j* on strain *i*. There are in total 71 promotion interactions with such a high confidence (shown in red, Fig. R5c). We will use them as the ground truth to check our inference results on promotion effects (i.e., positive signs, shown in red in Fig. R5b). It turns out that in total we inferred 13 wrong positive signs (marked as '×' in Fig. R5c), and missed 5 positive signs (marked as '?' in Fig. R5c). Therefore, our inference of positive signs has an accuracy of 74.65% (53/71), if we set the fold growth threshold 20 as the indication of promotion effect. Finally, we also observed that the accuracy on the inference generally increased by increasing this threshold (Fig. R5d).

Fig R1

Fig. R1 | In case the sign-pattern of the Jacobian matrix is not constant, we can still infer its zero. **a.** The temporal evolution of the abundance of each species. **b.** The sign of J_{22} is time varying, while the signs of other elements in the Jacobian matrix are constant. **c.** Inference of J_{12} . According to the position of $\mathbf{x}^{\{1,2\}} - \mathbf{x}^{\{1\}}$, the green line which is orthogonal to the red line cannot produce a zero entry for \mathbf{J}_1 , implying that $J_{11} \neq 0$ and $J_{12} \neq 0$. This is consistent with the ground truth. **d.** Inference of J_{12} . The green line (orthogonal to the blue line) is aligned with the x_2 -axis, indicating that $J_{21} = 0$ and $J_{22} \neq 0$, consistent with the ground truth. **e.** When noise level $\eta = 0.1$, the light green line is orthogonal to the light red line corresponding to the difference of two noisy samples $\mathbf{x}^{\{1,2\}}$ and $\mathbf{x}^{\{1\}}$. The bold red and green lines correspond to the noiseless case. There are in total 1000 different measurements (replicates). We find that the angles between the green lines and x_1 -axis are big enough, which help us conclude that $J_{12} \neq 0$ with high confidence. **f.** When noise level $\eta = 0.1$, the light green line is orthogonal to the light blue line corresponding to the difference of two noisy samples $\mathbf{x}^{\{1,2\}}$ and $\mathbf{x}^{\{2\}}$. The bold blue and green lines correspond to the noiseless case. There are in total 1000 replicates. Among the 1000 replicates, the light green line is equally distributed to the left and the right side of x_2 -axis, indicating that the deviation of the light green line from the x_2 -axis is likely due to measurement noises. This behavior let us introduce a user-defined cutoff value to judge the zero-pattern of J_{ij} based on the average deviation between the x_1 -axis (or x_2 -axis) and green lines.

Fig. R2

Fig. R2 | In case the sign-pattern of the Jacobian matrix is not constant, but the steady-state samples were still collected from the microbial community under the same or similar environmental conditions (e.g., nutrient availability), the results of our inference can be interpreted as the overall inhibition or promotion impact between different taxa. Here we consider a toy model of two species X and Y . Each has a per capita growth rate that is modulated by its mutualistic partner as well as the resource amount (denoted as a). The population dynamics model is shown in Eq. (R2), with model parameters $\kappa = 0.12, \delta = 0.5, \beta = 2$. **a.** Three regimes of the interaction types emerge from different resource amount, from mutualism, parasitism to competition. For a given a , comparing the difference of steady states between monocultures (dashed lines) and co-cultures (solid lines) can give the relative yields as the ground truth. Because that Jacobian may change the signs over time, the sign of relative yields can be interpreted as the overall (all effective) impact between two taxa, denoted as J_{eff} . **b-d.** Diagrams of our inference method under different resource amount. \tilde{J}_{eff} denotes as the inferred sign-pattern of overall impacts between two taxa. **e-g.** $J_{ij}(\mathbf{x}^l + \sigma(\mathbf{x}^k - \mathbf{x}^l))$ as a function of σ under different resource amount.

Fig. R3 | Inferring interaction types in a synthetic microbial community of two cross-feeding partners with different amount of resource availability [5]. a-c. The abundance of the cocultures (solid line) and monocultures (dashed line) for the Trp (green) and Leu (red) strains with the resources of low, medium and high amino acid. Leu and Trp strains are auxotrophic for each other. However, their co-cultures under different amount of resources exhibit different interaction types. **d-f.** Diagrams of our inference method. The inferred results are consistent with the experimental observations.

Fig. R4 | Inferring interaction types in a synthetic community of maize roots with 7 bacterial species [1]. **a.** The dataset consists of 7 sextets (i.e., 7 steady-state samples involving 6 of the 7 species) and 1 septet (i.e., a steady-state sample involving all the 7 species). **b.** We find that R^2 of all fitted hyperplanes are smaller than 0.9, implying that the given steady-state samples cannot be properly described by the GLV model. Hence, we should focus on the inference of interaction types, rather than interaction strengths. **c.** Inferred interaction types by our brute-force method only using the 7 sextets. **d.** The changes of species abundance before and after respectively adding one species into the 6-bacterial communities. Each column corresponds to a 6-bacterial community, the name of the newly introduced species is marked in the top of each column. Blue (or red) corresponds to the decrement (or increment) of one species after introducing a new species into sextets, respectively. 'x' indicates false prediction. There are in total 12 false predictions.

Fig. R5 | Inferring interaction types in a synthetic community of 14 strains each containing a gene knockout that lead to an auxotrophic phenotype of 1 of 14 essential amino acids [6]. The dataset consists of 91 steady-state samples, each involving a particular pair of the 14 strains. **a.** We find that R^2 of all fitted hyperplanes are smaller than 0.9, suggesting that this community cannot be properly described by the GLV model. **b.** Inferred interaction types by our heuristic algorithm (with 1000 user-defined intersection lines) using the 91 steady-state samples. **c.** The experimentally measured fold growth matrix $F = (F_{ij})$, with F_{ij} the fold growth of strain i (row) in the co-culture paired with strain j (column). We set $F_{ij} \geq 20$ as the indication of promotion effect of strain j on strain i . There are in total 71 promotion interactions with such a higher confidence (shown in red). Among them, 53 were correctly inferred, 13 (marked as ‘x’) were not correctly inferred, and 5 (marked as ‘?’) were undetermined by our method, resulting in accuracy $53/71 = 74.65\%$, at this particular fold growth threshold. **d.** The inference accuracy of promotion effect as a wide range of threshold value of fold growth.

References:

1. Niu, B., Paulson, J. N., Zheng, X. & Kolter, R. Simplified and representative bacterial community of maize roots. *Proceedings of the National Academy of Sciences* **114**, E2450–E2459 (2017).
2. Bucci, V. *et al.* MDSINE: Microbial Dynamical Systems INference Engine for microbiome time-series analyses. *Genome Biology* **17**, (2016).
3. Cao, H.-T., Gibson, T. E., Bashan, A. & Liu, Y.-Y. Inferring human microbial dynamics from temporal metagenomics data: Pitfalls and lessons. *BioEssays* **39**, 1600188 (2017).
4. Friedman, J., Higgins, L. M. & Gore, J. Community structure follows simple assembly rules in microbial microcosms. *Nature Ecology & Evolution* **1**, 109 (2017).
5. Hoek, T. A. *et al.* Resource Availability Modulates the Cooperative and Competitive Nature of a Microbial Cross-Feeding Mutualism. *PLOS Biology* **14**, e1002540 (2016).
6. Mee, M. T., Collins, J. J., Church, G. M. & Wang, H. H. Syntrophic exchange in synthetic microbial communities. *PNAS* **111**, E2149–E2156 (2014).
7. Kim, H. J., Boedicker, J. Q., Choi, J. W. & Ismagilov, R. F. Defined spatial structure stabilizes a synthetic multispecies bacterial community. *PNAS* **105**, 18188–18193 (2008).
8. Angulo, M. T., Moog, C. H. & Liu, Y.-Y. Controlling microbial communities: a theoretical framework. *bioRxiv* 149765 (2017). doi:10.1101/149765.

REVIEWERS' COMMENTS:

Reviewer #1 (Remarks to the Author):

The authors have addressed my primary concerns. The analysis of additional microbial systems strengthens the claims, particularly the analysis of a system in which signs change over experimental conditions. There remain constraints on the generality of this approach, however, these constraints can be evaluated by the field.

Reviewer #2 (Remarks to the Author):

The revised manuscript successfully addresses all my points. I recommend acceptance.

Reviewer #3 (Remarks to the Author):

I very much appreciated the level of care with which the authors revised their work, including the extra material in the already extensive SI. I hope the authors agree that these changes improved the manuscript, especially in regards to application to additional experimental datasets. Although the revisions have rendered the text a little harder to parse at points, overall it is more persuasive and I now find it much more plausible that this inference method will indeed find a broad application. I also appreciate that the potential limits of its applicability are clearly stated (and even emphasized).